# Structures and Bioactivities of Psolusosides B_1_, B_2_, J, K, L, M, N, O, P, and Q from the Sea Cucumber *Psolus fabricii*. The First Finding of Tetrasulfated Marine Low Molecular Weight Metabolites

**DOI:** 10.3390/md17110631

**Published:** 2019-11-06

**Authors:** Alexandra S. Silchenko, Anatoly I. Kalinovsky, Sergey A. Avilov, Vladimir I. Kalinin, Pelageya V. Andrijaschenko, Pavel S. Dmitrenok, Roman S. Popov, Ekaterina A. Chingizova

**Affiliations:** G.B. Elyakov Pacific Institute of Bioorganic Chemistry, Far Eastern Branch of the Russian Academy of Sciences, Pr. 100-letya Vladivostoka 159, Vladivostok 690022, Russia; sialexandra@mail.ru (A.S.S.); avilov-1957@mail.ru (S.A.A.); pandryashchenko@mail.ru (P.V.A.); paveldmt@piboc.dvo.ru (P.S.D.); rs.popov@outlook.com (R.S.P.); martyyas@mail.ru (E.A.C.)

**Keywords:** *Psolus fabricii*, triterpene glycosides, psolusosides, sea cucumber, cytotoxic activity

## Abstract

Ten new di-, tri- and tetrasulfated triterpene glycosides, psolusosides B_1_ (**1**), B_2_ (**2**), J (**3**), K (**4**), L (**5**), M (**6**), N (**7**), O (**8**), P (**9**), and Q (**10**), were isolated from the sea cucumber *Psolus fabricii* collected in the Sea of Okhotsk near the Kurile Islands. Structures of these glycosides were established by two-dimensional (2D) NMR spectroscopy and HR-ESI mass-spectrometry. It is particularly interesting that highly polar compounds **9** and **10** contain four sulfate groups in their carbohydrate moieties, including two sulfates in the same terminal glucose residue. Glycoside **2** has an unusual non-holostane aglycone with 18(16)-lactone and a unique 7,8-epoxy fragment. Cytotoxic activities of compounds **1**–**10** against several mouse cell lines such as Ehrlich ascites carcinoma cells, neuroblastoma Neuro 2A, normal epithelial JB-6 cells, and erythrocytes were quite different depending both on structural peculiarities of these glycosides and the type of cells subjected to their actions. Psolusoside L (**5**), pentaoside, with three sulfate groups at C-6 of two glucose and one 3-*O*-methylglucose residue and holostane aglycone, is the most active compound in the series. The presence of a sulfate group at C-2 of the terminal glucose residue attached to C-4 of the first (xylose) residue significantly decreases activities of the corresponding glycosides. Psolusosides of group B (**1**, **2**, and known psolusoside B) are inactive in all tests due to the presence of non-holostane aglycones and tetrasaccharide-branched sugar chains sulfated by C-2 of Glc4.

## 1. Introduction

Triterpene glycosides of sea cucumbers are well known by their structural diversity and promising biological effects [1,2,3], including cytotoxicity against cancer cells and antitumor activity [4,5,6]. Therefore, the search for new representatives of this class of marine natural products and studies of their biological activities seem to be relevant. Moreover, structural analysis of diverse glycosides of sea cucumbers allows us to understand the peculiarities of biosynthesis of these complicated and numerous marine metabolites.

As a continuation of our investigation of triterpene glycoside composition of the sea cucumber *Psolus fabricii* (Psolidae, Dendrochirotida) [7,8,9,10,11,12] we report herein the isolation of ten new glycosides, psolusosides B_1_ (**1**), B_2_ (**2**), J–Q (**3**–**10**), and their structural elucidation based on the ^1^H, ^13^C NMR, one-dimensional (1D) TOCSY and 2D NMR (^1^H,^1^H-COSY, HMBC, HSQC, ROESY), and HR-ESI mass spectrometry. The hemolytic activities against mouse erythrocytes and cytotoxic activities of **1**–**10** against mouse Ehrlich ascites carcinoma cells, neuroblastoma Neuro 2A, and normal epithelial JB-6 cells have been studied.

## 2. Results and Discussion

### 2.1. Structural Elucidation of the Glycosides

The initial stages of isolation of compounds **1**–**10** were the same as for other glycosides from *P. fabricii* and were described earlier [10,11,12]. The individual glycosides were isolated by HPLC on reversed-phase columns to give psolusosides: B_1_ (**1**) (7,3 mg), B_2_ (**2**) (3.4 mg), J (**3**) (4.8 mg), K (**4**) (3.4 mg), L (**5**) (60 mg), M (**6**) (1.0 mg), N (**7**) (8.8 mg), O (**8**) (0.6 mg), P (**9**) (8.5 mg), and Q (**10**) (1.4 mg) (Figure 1).

The ^1^H and ^13^C NMR spectra corresponding to the carbohydrate chains of psolusosides B_1_ (**1**) and B_2_ (**2**) were coincident to each other and to those of known psolusoside B [12] showing the identity of their tetrasaccharide carbohydrate moieties branched by C-4 of the xylose unit and having two sulfate groups (Appendix A).

The molecular formula of psolusoside B_1_ (**1**) was determined to be C_55_H_82_O_31_S_2_Na_2_ from the [M_2Na_ − Na]**^−^** ion peak at *m/z* 1325.4164 (calc. 1325.4185) and [M_2Na_ − 2Na]^2**−**^ ion peak at *m/z* 651.2157 (calc. 651.2146) in the (−)HR-ESI-MS. The signal of H-16 was observed as a broad singlet at δ_H_ 4.89 and the signal of H-17 was observed as a singlet at δ_H_ 2.97 in the ^1^H NMR spectrum of **1**. These data as well as corresponding signals of carbons at δ_C_ 79.9 (C-16) and δ_C_ 58.8 (C-17) (Table 1) were indicative for 18(16)-lactone moiety (Table 1). *O*-acetyl group (δ_C_ 170.9 (CH_3_COO) and 21.6 (CH_3_COO) in the ^13^C NMR spectrum), attached to C-20, caused the deshielding of its signal to δ_C_ 83.8 in the same manner as in the spectrum of psolusoside B [12]. The side chain of **1** was identical to that of psolusoside B due to the coincidence of those signals in the ^1^H and ^13^C NMR spectra. The signal at δ_C_ 199.3 corresponded to a keto-group adjacent to a double bond (the signals of olefinic carbons at δ_C_ 135.3 (C-8) and 169.0 (C-9)). The position of the keto-group was deduced as C-7 based on the correlations between H_2_-6 (δ_H_ 2.42 and δ_H_ 2.29) and C-7 (δ_C_ 199.3) in the HMBC spectrum of **1**. This was also corroborated by an isolated spin system between the doublet of doublets at δ_H_ 1.54 (H-5) and another doublet of doublets at δ_H_ 2.42 (H-6a) and the triplet at δ_H_ 2.29 (H-6b) observed in the ^1^H,^1^H-COSY spectrum. The 8(9)-position of double bond was confirmed by the HMBC correlations H_3_-32/C-8 and H_3_-19/C-9. So, the aglycone of psolusoside B_1_ (**1**) is characterized by the unique combination of such structural features as 7-keto-8(9)-ene fragment and 18(16)-lactone.

The (−)ESI-MS/MS of **1** demonstrated the fragmentation of [M_2Na_ − Na]^−^ ion at *m/z* 1325.4. The peaks of fragment ions were observed at *m/z* 1265.4 [M_2Na_ − Na − CH_3_COOH]^−^, 1145.4 [M_2Na_ − Na − CH_3_COOH − NaHSO_4_]^−^, 1001.4 [M_2Na_ − Na − CH_3_COOH − C_6_H_10_O_8_SNa (GlcSO_3_Na) + H]^−^, and 839.3 [M_2Na_ − Na − CH_3_COOH − GlcSO_3_Na − Glc + H]^−^ corroborating the structure of psolusoside B_1_ (**1**).

All these data indicate that psolusoside B_1_ (**1**) is 3*β*-*O*-{6-*O*-sodium-sulfate-*β*-d-glucopyranosyl-(1→4)-*β*-d-glucopyranosyl-(1→2)-[2-*O*-sodium-sulfate-*β*-d-glucopyranosyl-(1→4)]-*β*-d-xylopyranosyl}-7-keto-20(*S*)-acetoxylanosta-8,25-diene-18(16)-lactone.

The molecular formula of psolusoside B_2_ (**2**) (C_55_H_82_O_31_S_2_Na_2_) was determined to be the same as of **1** from the [M_2Na_ − Na]^−^ ion peak at *m/z* 1325.4163 (calc. 1325.4185) and [M_2Na_ − 2Na]^2−^ ion peak at *m/z* 651.2159 (calc. 651.2146) in the (−)HR-ESI-MS. In the ^1^H and ^13^C NMR spectra of the aglycone part of **2** the signals characteristic of 18(16)-lactone (δ_H_ 4.94 (brs, H-16), δ_H_ 3.01 (s, H-17), δ_C_ 79.3 (C-16), and δ_C_ 60.2 (C-17) as well as *O*-acetylated C-20 (δ_H_ 2.05 (s, CH**_3_**COO), δ_C_ 21.8 (CH_3_COO), δ_C_ 170.9(CH_3_COO), and δ_C_ 83.9 (C-20)) were observed (Table 2). The side chains in aglycones of **1** and **2** were identical to each other. The signal at δ_H_ 3.10 (d, 6.6, H-7) was assigned by the ^1^H,^1^H-COSY spectrum where the protons H-5/H-6/H-7 formed an isolated spin system. The corresponding signal of C-7 at δ_C_ 56.2 was deduced by the HSQC spectrum of **2**. The signal of quaternary C-8 assigned by the HMBC correlations H_3_-32/C-8, H_2_-6/C-8, and H-7/C-8 was deshielded to δ_C_ 59.6 in the ^13^C NMR spectrum. These data indicated the presence of an oxygen-bearing substituent at C-7 and C-8, which was supposed to be an 7,8-epoxide [13] that correlated with the MS data. The olefinic broad doublet of doublets at δ_H_ 6.00 was assigned to H-11 due to its correlation with H_2_-12 (δ_H_ 2.81 (dd, 5.2; 17.6, H-12a) and 2.60 (brdd, 2.3; 17.6, H-12b)) in the ^1^H,^1^H-COSY spectrum. The signal at δ_C_ 122.7 corresponded to olefinic C-11 and was deduced by the HSQC spectrum. So, the double bond could occupy the 9(11)-position only. The signal of C-9 at δ_C_ 143.2 correlated in the HMBC spectrum with both δ_H_ 2.81 (H-12a) and 2.60 (H-12b) and the methyl singlet δ_H_ 1.13 (H_3_-19).

The configuration of C-7 was established as (*S*) by the ROE-correlation H-7/H_3_-32 and was confirmed by the coupling pattern of H-7 (δ_H_ 3.10 (d, 6.6)), that coincided with the calculated coupling constant based on dihedral angle values in the optimized MM2 model of aglycone of psolusoside B_2_ (**2**) having H-7*α*-orientation and 8(*R*)-configuration. Thus, the aglycone of psolusoside B_2_ (**2**) has unprecedented 7(*S*),8(*R*)-epoxy-20(*S*)-acetoxylanosta-9(11),25-diene-18(16)-lactone structure.

The (−)ESI-MS/MS of **2** showed the fragmentation of [M_2Na_ − Na]^−^ ion at *m/z* 1325.4. The peaks of fragment ions were observed at the same *m/z* values of 1265.4, 1145.4, 1001.4, and 839.3 as in the spectrum of **1**, corroborating the identity of the carbohydrate chains of **1** and **2**. Additionally, the fragment ion-peaks at *m/z* 535.1 [M_2Na_ − Na − C_32_H_45_O_6_ (Agl) − C_6_H_10_O_8_SNa (GlcSO_3_Na)]^−^ and 403 [M_2Na_ − Na − C_32_H_45_O_6_ (Agl) − C_6_H_10_O_8_SNa (GlcSO_3_Na) − Xyl (C_5_H_8_O_4_)]^−^ corresponding to the tri- and disaccharide fragments, were observed in the MS/MS spectrum of **2**.

All these data indicate that psolusoside B_2_ (**2**) is 3*β*-*O*-{6-*O*-sodium-sulfate-*β*-d-glucopyranosyl-(1→4)-*β*-d-glucopyranosyl-(1→2)-[2-*O*-sodium-sulfate-*β*-d-glucopyranosyl-(1→4)]-*β*-d-xylopyranosyl}-7(*S*),8(*R*)-epoxy-20(*S*)-acetoxylanosta-9(11),25-diene-18(16)-lactone.

The molecular formula of psolusoside J (**3**) (C_53_H_79_O_32_S_3_Na_3_) was determined from the [M_3Na_ − Na]**^−^** ion peak at *m/z* 1369.3485 (calc. 1369.3517), [M_3Na_ − 2Na]^2**−**^ ion peak at *m/z* 673.1812 (calc. 673.1813), and [M_3Na_ − 3Na]^3**−**^ ion peak at *m/z* 441.1248 (calc. 441.1244) in the (−)HR-ESI-MS. The ^1^H and ^13^C NMR spectra of the aglycone part of psolusoside J (**3**) coincided with those of psolusoside H isolated earlier from *P. farbricii* [12] (Appendix A) indicating the identity of their holostane-type aglycones having 7(8)- and 25(26)-double bonds and 16-keto-group. This aglycone is common for the glycosides of sea cucumbers belonging to the orders Dendrochirotida and Aspidochirotida [2,12].

In the ^1^H and ^13^C NMR spectra of the carbohydrate part of psolusoside J (**3**) four characteristic doublets at δ_H_ 4.60–5.12 (*J* = 7.3 − 8.1 Hz) and, corresponding to them, signals of anomeric carbons at δ_C_ 101.7–105.5 were indicative of a tetrasaccharide chain and *β*-configurations of glycosidic bonds. The ^13^C NMR spectra of tetrasaccharide carbohydrate chain of **3** and those of **1** and **2** were quite different, while the ^1^H,^1^H-COSY and 1D TOCSY spectra of **3** showed the signals of four isolated spin systems assigned to one xylose and three glucose residues as in psolusosides B [12], B_1_ (**1**), and B_2_ (**2**). The positions of interglycosidic linkages were elucidated by the ROESY and HMBC spectra of **3** (Table 3), where the correlations between H-1 of the xylose (Xyl1) and H-3 (C-3) of the aglycone, H-1 of the second residue (glucose, Glc2) and H-2 (C-2) of the xylose (Xyl1), H-1 of the third residue (glucose, Glc3) and H-4 (C-4) of the second residue (glucose, Glc2), H-1 of the fourth residue (glucose, Glc4) and H-4 (C-4) of the first residue (xylose, Xyl1) were observed, indicating the same architecture of sugar chains in **3** and **1** and **2**. The comparison of the NMR spectra of **1** and **3** showed the coincidence of the signals of three monosaccharide residues corresponding to the linear part of the carbohydrate chain (residues I–III). The signals of terminal monosaccharide unit attached to C-4 of the first (Xyl1) unit, assigned by the ^1^H,^1^H-COSY and 1D TOCSY spectra of **3** were indicative of a sulfated by C-2 glucose residue due to characteristic shifting effects observed in the ^13^C NMR spectrum: the signal of C-2 Glc4 was deshielded to δ_C_ 81.2 and the signal of C-1 Glc4 was shielded to δ_C_ 101.7 in comparison with the corresponding signals of the same sugar unit in the ^13^C NMR spectrum of psolusoside I isolated by us earlier [12].

The δ_C_ of the signals of C-2 and C-1 of the fourth monosaccharide unit (Glc4) in the ^13^C NMR spectrum of psolusoside J (**3**) were very close to those in the ^13^C NMR spectrum of **1**, corroborating the presence of a sulfate group at C-2 of this residue (Glc4). The correlations between H-2/H-3/H-4 in this monosaccharide residue, deduced by the ^1^H,^1^H-COSY spectrum of **3**, indicated the signal of H-4 Glc4 at δ_H_ 4.90. The signal of the corresponding carbon (C-4 Glc4), deduced by the HSQC spectrum, was downshifted to δ_C_ 77.3 as compared with the same signal (C-4 Glc4) at δ_C_ 70.7 in the ^13^C NMR spectrum of **1**. Actually, the signals at δ_C_ ~70.4–70.8 were absent and the signals of C-3 Glc4 and C-5 Glc4 were upshifted to δ_C_ 75.6 and 76.6, correspondingly, in the ^13^C NMR spectrum of **3** due to β-shifting effect of sulfate group, when compared with the corresponding signals in the ^13^C NMR spectrum of **1**. Considering that (−)HR-ESI-MS indicated the presence of three sulfate groups as well as the NMR data, the attachment of the third sulfate group to C-4 of Glc4 was supposed. The signal at δ_C_ 62.4 (C-6 Glc4) was characteristic for carbons of non-sulfated hydroxy-methylene groups of glucopyranose residues and excluded the positioning of the third sulfate group at C-6 Glc4 that confirmed our supposition. Hence psolusoside J (**3**) is a trisulfated tetraoside with two sulfate groups attached to the same glucose residue. To the best our knowledge, this structural feature is first found in the glycosides.

The (−)ESI-MS/MS of **3** demonstrated the fragmentation of [M_3Na_ − Na]^−^ ion at *m/z* 1369.3. The peaks of fragment ions were observed at *m/z* 1249.4 [M_3Na_ − Na − NaHSO_4_]^−^, 1105.4 [M_3Na_ − Na − C_6_H_9_O_8_SNa (GlcSO_3_Na)]^−^, 1003.4 [M_3Na_ − Na − C_6_H_9_O_8_SNa (GlcSO_3_Na) − NaSO_3_ + H]^−^, 841.4 [M_3Na_ − Na − NaSO_3_ − GlcSO_3_Na − Glc + H]^−^, 403.0 [M_3Na_ − Na − C_30_H_43_O_4_ (Agl) − C_6_H_9_O_11_S_2_Na_2_ (Glc(SO_3_Na)_2_) − Xyl (C_5_H_8_O_4_)]^−^, and 241.0 [M_3Na_ − Na − C_30_H_43_O_4_ (Agl) − C_6_H_9_O_11_S_2_Na_2_ (Glc(SO_3_Na)_2_) − Xyl (C_5_H_8_O_4_) − Glc (C_6_H_10_O_5_)]^−^, corroborating the structure of psolusoside J (**3**).

All these data indicate that psolusoside J (**3**) is 3*β*-*O*-{6-*O*-sodium-sulfate-*β*-d-glucopyranosyl-(1→4)-*β*-d-glucopyranosyl-(1→2)-[2,4-*O*-sodium-disulfate-*β*-d-glucopyranosyl-(1→4)]-*β*-d-xylopyranosyl}-16-ketoholosta-7,25-diene.

The ^13^C NMR spectra of the aglycone moieties of the glycosides **4**–**10** were identical to each other (Appendix A) and to those of psolusosides E, F, and G containing 16-ketoholosta-9(11),25-dien-3β-ol as an aglycone, known earlier and frequently occurring in the glycosides of sea cucumbers [12].

The molecular formula of psolusoside K (**4**) was determined to be C_53_H_79_O_32_S_3_Na_3_ from the [M_3Na_ − Na]**^−^** ion peak at *m/z* 1369.3485 (calc. 1369.3517), [M_3Na_ − 2Na]^2**−**^ ion peak at *m/z* 673.1821 (calc. 673.1813), and [M_3Na_ − 3Na]^3**−**^ ion peak at *m/z* 441.1255 (calc. 441.1244) in the (−)HR-ESI-MS and was coincident with the formula of psolusoside J (**3**). In the ^1^H and ^13^C NMR spectra of the carbohydrate moiety of psolusoside K (**4**) four characteristic doublets at δ_H_ 4.61–5.07 (*J* = 7.2–8.4 Hz) and corresponding signals of anomeric carbons at δ_C_ 101.4–104.7 were indicative of a tetrasaccharide chain and *β*-configurations of glycosidic bonds. The positions of interglycosidic linkages were elucidated by the ROESY and HMBC spectra of **4** (Table 4) as described above indicating the presence of a tetrasaccharide carbohydrate chain branched by C-4 of the xylose residue (Xyl1). The monosaccharide composition of **4**, deduced from the ^1^H,^1^H-COSY and 1D TOCSY spectra, was the same as in glycosides **1**–**3**. The comparison of the ^13^C NMR spectra of trisulfated compounds **3** and **4** showed the coincidence of the signals corresponding to three monosaccharide residues (residues I–III in the formula) forming the linear part of the sugar chain. The signals of C-2 Glc4 at δ_C_ 80.3 and C-1 Glc4 at δ_C_ 101.4 in the ^13^C NMR spectrum of **4** were very close to those in the spectrum of **3** that indicated the attachment of a sulfate group to C-2 Glc4 in psolusoside K (**4**). All of the signals of this monosaccharide residue were assigned using the ^1^H,^1^H-COSY and 1D TOCSY spectra. The doublet at δ_H_ 5.00 and the doublet of doublets at δ_H_ 4.63 corresponded to the protons of the hydroxy-methylene group of the terminal glucose unit (H_2_-6 Glc4) and were deshielded as compared with the corresponding signals in the ^1^H NMR spectrum of **3**. The signal at δ_C_ 67.4 (C-6 Glc4) also indicated the presence of a sulfate group at C-6 of Glc4 in addition to another sulfate group at C-2 of Glc4. So, psolusoside K (**4**) is an isomer of psolusoside J (**3**) by the sulfate position and is the second glycoside from sea cucumbers that contains two sulfate groups bonded to the same monosaccharide residue.

The (−)ESI-MS/MS of **4** demonstrated the fragmentation of [M_3Na_ − Na]^−^ ion at *m/z* 1369.3. The peaks of fragment ions were observed at the same *m/z*: 1249.4 [M_3Na_ − Na − NaHSO_4_]^−^, 1105.4 [M_3Na_ − Na − C_6_H_9_O_8_SNa (GlcSO_3_Na)]^−^, 1003.4 [M_3Na_ − Na − C_6_H_9_O_8_SNa (GlcSO_3_Na) − NaSO_3_ + H]^−^, 403.0 [M_3Na_ − Na − C_30_H_43_O_4_ (Agl) − C_6_H_9_O_11_S_2_Na_2_ (Glc(SO_3_Na)_2_) − Xyl (C_5_H_8_O_4_)]^−^, and 241.0 [M_3Na_ − Na − C_30_H_43_O_4_ (Agl) − C_6_H_9_O_11_S_2_Na_2_ (Glc(SO_3_Na)_2_) − Xyl (C_5_H_8_O_4_) − Glc (C_6_H_10_O_5_)]^−^ as in the MS/MS of psolusoside J (**3**) corroborating their isomerism.

All these data indicate that psolusoside K (**4**) is 3*β*-*O*-{6-*O*-sodium-sulfate-*β*-d-glucopyranosyl-(1→4)-*β*-d-glucopyranosyl-(1→2)-[2,6-*O*-sodium-disulfate-*β*-d-glucopyranosyl-(1→4)]-*β*-d-xylopyranosyl}-16-ketoholosta-9,25-diene.

The molecular formula of psolusoside L (**5**) (C_60_H_91_O_36_S_3_Na_3_) was determined from the [M_3Na_ − Na]**^−^** ion peak at *m/z* 1529.4222 (calc. 1529.4253), [M_3Na_ − 2Na]^2**−**^ ion peak at *m/z* 753.2190 (calc. 753.2180), and [M_3Na_ − 3Na]^3**−**^ ion peak at *m/z* 494.4835 (calc. 494.4823) in the (−)HR-ESI-MS, indicating the presence of three sulfate groups. In the ^1^H and ^13^C NMR spectra of the carbohydrate part of psolusoside L (**5**) five characteristic doublets at δ_H_ 4.65–5.16 (*J* = 6.9–8.1 Hz) and, corresponding to them, signals of anomeric carbons at δ_C_ 103.4–104.8 were indicative of a pentasaccharide chain and *β*-configurations of glycosidic bonds (Table 5). Analysis of the ^1^H,^1^H-COSY and 1D TOCSY spectra of psolusoside L (**5**) showed the presence of one xylose, one quinovose, two glucose, and one 3-*O*-methylglucose residues. The presence of a quinovose residue was confirmed by the ^1^H and ^13^C NMR spectra demonstrating the characteristic doublet at δ_H_ 1.59 (H-6 Qui2) and the signal at δ_C_ 17.7 (C-6 Qui2). The positions of interglycosidic linkages and the consequence of monosaccharides in the chain of **5** were established by analysis of the ROESY and HMBC spectra (Table 5) indicating the presence of branched pentasaccharide moiety with glucose, attached to C-4 Xyl1, and 3-*O*-methylglucose, attached to C-3 Glc3, as terminal residues. The ^13^C NMR spectrum of **5** demonstrated three signals at δ_C_ 67.0, 67.5, and 67.6, corresponding to sulfated hydroxy-methylene groups of glucopyranose residues that indicated the sulfation of two glucose and 3-O-methylglucose units in the carbohydrate chain of **5**.

The comparison of the ^13^C NMR spectrum of the sugar part of psolusoside L (**5**) with those of known achlioniceosides A_1_, A_2_, and A_3_, with identical carbohydrate chains, isolated earlier from the sea cucumber *Rhipidothuria racowitzai* [14] showed the coincidence of the signals of four monosaccharide residues in their spectra. The signals of terminal 3-*O*-methylglucose residues of the novel and known compounds were different due to the absence of a sulfate group in this residue of known compounds. All these data indicated that psolusoside L (**5**) is a pentaoside with a new trisulfated carbohydrate chain branched by C-4 Xyl1.

The (−)ESI-MS/MS of **5** demonstrated the fragmentation of [M_3Na_ − Na]^−^ ion at *m/z* 1529.4. The peaks of fragment ions were observed at *m/z*: 1409.5 [M_3Na_ − Na − NaHSO_4_]^−^, 1265.4 [M_3Na_ − Na − C_6_H_9_O_8_SNa (GlcSO_3_Na)]^−^, 1131.5 [M_3Na_ − Na − C_7_H_12_O_9_SNa (MeGlcSO_3_Na) − NaSO_3_]^−^, 665.1 [M_3Na_ − Na − C_30_H_43_O_4_ (Agl) − C_7_H_12_O_9_SNa (MeGlcSO_3_Na) − NaSO_3_]^−^, and 519.0 [M_3Na_ − Na − C_30_H_43_O_4_ (Agl) − C_7_H_12_O_9_SNa (MeGlcSO_3_Na) − C_6_H_9_O_7_SNa (GlcSO_3_Na)]^−^, confirming the structure of psolusoside L (**5**).

All these data indicate that psolusoside L (**5**) is 3*β*-*O*-{6-*O*-sodium-sulfate-3-*O*-methyl-*β*-d-glucopyranosyl-(1→3)-6-*O*-sodium-sulfate-*β*-d-glucopyranosyl-(1→4)-*β*-d-quinovopyranosyl-(1→2)-[6-*O*-sodium-sulfate-*β*-d-glucopyranosyl-(1→4)]-*β*-d-xylopyranosyl}-16-ketoholosta-9(11),25-diene.

The molecular formula of psolusoside M (**6**) was determined to be C_60_H_91_O_36_S_3_Na_3_ from the ion peaks at *m/z* 1529.4273 (calc. 1529.4253) [M_3Na_ − Na]**^−^**, 753.2202 (calc. 753.2180) [M_3Na_ − 2Na]^2**−**^, and 494.4844 (calc. 494.4823) [M_3Na_ − 3Na]^3**−**^ in the (−)HR-ESI-MS, indicating this glycoside to be an isomer of psolusoside L (**5**). In the ^1^H and ^13^C NMR spectra of the carbohydrate part of psolusoside M (**6**) five characteristic doublets at δ_H_ 4.58–5.15 (*J* = 7.1–8.5 Hz) and, corresponding to them, signals of anomeric carbons at δ_C_ 100.9–104.8, were indicative of a pentasaccharide chain and *β*-configurations of glycosidic bonds (Table 6). Analysis of the ^1^H,^1^H-COSY and 1D TOCSY, ROESY, and HMBC spectra of psolusoside M (**6**) showed the same monosaccharide composition and architecture of the carbohydrate chain as in **5**. Actually, the comparison of their ^13^C NMR spectra showed the closeness of the signals corresponding to the monosaccharides from the first to the fourth. The differences of the ^13^C NMR spectra of compounds **6** and **5** were concerned with the terminal glucose residue (Glc5) connected to C-4 Xyl1. The characteristic signals at δ_C_ 100.9 (C-1 Glc5) and at δ_C_ 80.6 (C-2 Glc5) in the ^13^C NMR spectrum of **6** were very close to the corresponding signals in the spectra of the compounds **1**–**4** indicating the presence of a sulfate group at C-2 Glc5 in the psolusoside M (**6**). At the same time, the hydroxy-methylene group of this sugar was free from sulfation, since the signal of C-6 Glc5 was observed at δ_C_ 61.8. Two signals of sulfated hydroxy-methylene groups of the glucose (Glc3) and 3-*O*-methylglucose (MeGlc4) residues were observed at δ_C_ 67.5 and 67.0 in the ^13^C NMR spectrum of **6**. Therefore, psolusoside M (**6**) is an isomer of psolusoside L (**5**) by the sulfate group position.

The (−)ESI-MS/MS of **6** demonstrated the fragmentation of [M_3Na_ − Na]^−^ ion at *m/z* 1529.4. The peaks of fragment ions were observed at the same *m/z*: 1409.5 [M_3Na_ − Na − NaHSO_4_]^−^, 1265.4 [M_3Na_ − Na − C_6_H_9_O_8_SNa (GlcSO_3_Na)]^−^, 1131.5 [M_3Na_ − Na − C_7_H_12_O_9_SNa (MeGlcSO_3_Na) − NaSO_3_]^−^, 665.1 [M_3Na_ − Na − C_30_H_43_O_4_ (Agl) − C_7_H_12_O_9_SNa (MeGlcSO_3_Na) − NaSO_3_]^−^, and 519.0 [M_3Na_ − Na − C_30_H_43_O_4_ (Agl) − C_7_H_12_O_9_SNa (MeGlcSO_3_Na) − C_6_H_9_O_7_SNa (GlcSO_3_Na)]^−^ as in the MS/MS spectrum of glycoside **5**.

All these data indicate that psolusoside M (**6**) is 3*β*-*O*-{6-*O*-sodium-sulfate-3-*O*-methyl-*β*-d-glucopyranosyl-(1→3)-6-*O*-sodium-sulfate-*β*-d-glucopyranosyl-(1→4)-*β*-d-quinovopyranosyl-(1→2)-[2-*O*-sodium-sulfate-*β*-d-glucopyranosyl-(1→4)]-*β*-d-xylopyranosyl}-16-ketoholosta-9(11),25-diene.

The molecular formula of psolusoside N (**7**) was determined to be C_60_H_91_O_37_S_3_Na_3_ from the [M_3Na_ − Na]**^−^** ion peak at *m/z* 1545.4171 (calc. 1545.4202), [M_3Na_ − 2Na]^2**−**^ ion peak at *m/z* 761.2164 (calc. 761.2155), and [M_3Na_ − 3Na]^3**−**^ ion peak at *m/z* 499.8151 (calc. 499.8139) in the (−)HR-ESI-MS. In the ^1^H and ^13^C NMR spectra of the carbohydrate part of psolusoside N (**7**) five characteristic doublets at δ_H_ 4.67–5.12 (*J* = 6.8–8.3 Hz) and, corresponding to them, signals of anomeric carbons at δ_C_ 103.3–104.7 were indicative of a pentasaccharide chain and *β*-configurations of glycosidic bonds (Table 7). Analysis of the ^1^H,^1^H-COSY and 1D TOCSY spectra of psolusoside N (**7**) showed the presence of one xylose, three glucose, and one 3-*O*-methylglucose residues. The positions of interglycosidic linkages and the consequence of monosaccharides in the carbohydrate chain of **7** were established in the same manner as for **1**–**6** (Table 7) indicating the presence of branched pentasaccharide moiety having the same architecture as in compounds **5** and **6**. The comparison of the ^13^C NMR spectra of **7** and **5** showed the closeness of the signals of all the monosaccharide residues except for the signals assigned to the second sugar units in their chains. Actually, in the ^1^H and ^13^C NMR spectra of **7**, the signals characteristic of quinovose residue were absent but two doublets of doublets at δ_H_ 4.95 (H-6a Glc2) and at δ_H_ 4.75 (H-6b Glc2) and the signal at δ_C_ 61.0 (C-6 Glc2), assigned to hydroxy-methylene group of glucopyranose moiety, were detected. These data indicated the replacement of quinovose by the glucose residue in the second position of a carbohydrate chain in psolusoside N (**7**) as compared with psolusoside L (**5**). Three sulfate groups were supposed to attach the C-6 of two glucose and 3-*O*-methylglucose residues due to the signals at δ_C_ 67.4, 67.5, and 66.9 observed in the spectrum of **7**. The carbohydrate chain of psolusoside N (**7**) is the first found in the glycosides from holothurians.

The (−)ESI-MS/MS of **7** demonstrated the fragmentation of [M_3Na_ − Na]^−^ ion at *m/z* 1545.4. The peaks of fragment ions were observed at *m/z*: 1425.5 [M_3Na_ − Na − NaHSO_4_]^−^, 1281.4 [M_3Na_ − Na − C_6_H_9_O_8_SNa (GlcSO_3_Na)]^−^, 1147.5 [M_3Na_ − Na − C_7_H_12_O_9_SNa (MeGlcSO_3_Na) − NaSO_3_]^−^, 1003.4 [M_3Na_ − Na − C_6_H_9_O_8_SNa (GlcSO_3_Na) − C_7_H_12_O_8_SNa (MeGlcSO_3_Na) + H]^−^, 681.1 [M_3Na_ − Na − C_30_H_43_O_4_ (Agl) − C_7_H_12_O_9_SNa (MeGlcSO_3_Na) − NaSO_3_ + H]^−^, and 519.0 [M_3Na_ − Na − C_30_H_43_O_4_ (Agl) − C_7_H_12_O_9_SNa (MeGlcSO_3_Na) − C_6_H_9_O_7_SNa (GlcSO_3_Na)]^−^, corroborating the structure of psolusoside N (**7**).

All these data indicate that psolusoside N (**7**) is 3*β*-*O*-{6-*O*-sodium-sulfate-3-*O*-methyl-*β*-d-glucopyranosyl-(1→3)-6-*O*-sodium-sulfate-*β*-d-glucopyranosyl-(1→4)-*β*-d-glucopyranosyl-(1→2)-[6-*O*-sodium-sulfate-*β*-d-glucopyranosyl-(1→4)]-*β*-d-xylopyranosyl}-16-ketoholosta-9(11),25-diene.

The molecular formula of psolusoside O (**8**) was established as the same (C_60_H_91_O_37_S_3_Na_3_) as compound **7** from the [M_3Na_ − Na]**^−^** ion peak at *m/z* 1545.4197 (calc. 1545.4202), [M_3Na_ − 2Na]^2**−**^ ion peak at *m/z* 761.2171 (calc. 761.2155), and [M_3Na_ − 3Na]^3**−**^ ion peak at *m/z* 499.8155 (calc. 499.8139) in the (−)HR-ESI-MS.

In the ^1^H and ^13^C NMR spectra of the carbohydrate part of psolusoside O (**8**), five characteristic doublets at δ_H_ 4.60–5.12 (*J* = 7.0–8.6 Hz) and, corresponding to them, signals of anomeric carbons at δ_C_ 101.0–104.8 indicated a pentasaccharide carbohydrate chain and *β*-configurations of glycosidic bonds (Table 8).

Analysis of the ^1^H,^1^H-COSY and 1D TOCSY spectra of psolusoside O (**8**) showed the same monosaccharide composition and positions of interglycosidic linkages as in the carbohydrate chain of compound **7** (Table 8). The coincidence of the molecular formulae of **8** and **7** and the presence of three-charged ions in the (−)HR-ESI-MS of **8** indicated their difference in the position of a sulfate group. Really, the signals of monosaccharide residues from the first to the fourth were almost coincident in their ^13^C NMR spectra. The characteristic signals at δ_C_ 101.0 and δ_C_ 80.6 indicated the bonding of a sulfate group to C-2 of a terminal residue which glycosylates C-4 Xyl1. Analysis of the ^1^H,^1^H-COSY and 1D TOCSY spectra of **8** showed this unit is a glucose (Glc5). Indeed, the comparison of the ^13^C NMR spectra of **8** and **6** revealed their difference only in the signals of the second monosaccharide unit and the coincidence of the signals of the remaining ones. All these data indicate psolusoside O (**8**) has new trisulfated carbohydrate chain with the sulfate groups attached to C-6 of the third (Glc3), to C-6 of the fourth (MeGlc), and to C-2 of the fifth (Glc5) monosaccharide residues.

The (−)ESI-MS/MS of **8** demonstrated the fragmentation of [M_3Na_ − Na]^−^ with peaks of fragment ions, observed at *m/z* 1425.5 [M_3Na_ − Na − NaHSO_4_]^−^, 1281.4 [M_3Na_ − Na − C_6_H_9_O_8_SNa (GlcSO_3_Na)]^−^, 1161.5 [M_3Na_ − Na − C_6_H_9_O_8_SNa (GlcSO_3_Na) − NaHSO_4_]^−^, 1147.5 [M_3Na_ − Na − C_7_H_12_O_9_SNa (MeGlcSO_3_Na) − NaSO_3_]^−^, 1003.4 [M_3Na_ − Na − C_6_H_9_O_8_SNa (GlcSO_3_Na) − C_7_H_12_O_8_SNa (MeGlcSO_3_Na) + H]^−^, and 681.1 [M_3Na_ − Na − C_30_H_43_O_4_ (Agl) − C_7_H_12_O_9_SNa (MeGlcSO_3_Na) − NaSO_3_ + H]^−^, 519.0 [M_3Na_ − Na − C_30_H_43_O_4_ (Agl) − C_7_H_12_O_9_SNa (MeGlcSO_3_Na) − C_6_H_9_O_7_SNa (GlcSO_3_Na)]^−^, corroborating the isomerism of psolusosides O (**8**) and N (**7**).

All these data indicate that psolusoside O (**8**) is 3*β*-*O*-{6-*O*-sodium-sulfate-3-*O*-methyl-*β*-d-glucopyranosyl-(1→3)-6-*O*-sodium-sulfate-*β*-d-glucopyranosyl-(1→4)-*β*-d-glucopyranosyl-(1→2)-[2-*O*-sodium-sulfate-*β*-d-glucopyranosyl-(1→4)]-*β*-d-xylopyranosyl}-16-ketoholosta-9(11),25-diene.

The molecular formula of psolusoside P (**9**) was determined to be C_60_H_90_O_39_S_4_Na_4_ from the [M_4Na_ − Na]**^−^** ion peak at *m/z* 1631.3598 (calc. 1631.3641), [M_4Na_ − 2Na]^2**−**^ ion peak at *m/z* 804.1879 (calc. 804.1874), [M_4Na_ − 3Na]^3**−**^ ion peak at *m/z* 528.4628 (calc. 528.4619), and [M_4Na_ − 4Na]^4**−**^ ion peak at *m/z* 390.6001 (calc. 390.5991) in the (−)HR-ESI-MS indicating the presence of four sulfate groups. In the ^1^H and ^13^C NMR spectra of the carbohydrate part of psolusoside P (**9**), five characteristic doublets at δ_H_ 4.66–5.16 (*J* = 7.2–8.3 Hz) and corresponding signals of anomeric carbons at δ_C_ 103.1–104.8 were indicative of a pentasaccharide chain and *β*-configurations of glycosidic bonds (Table 9). Analysis of the ^1^H,^1^H-COSY and 1D TOCSY spectra of psolusoside P (**9**) showed the presence of one xylose, one quinovose, two glucose, and one 3-*O*-methylglucose residues. The positions of interglycosidic linkages and the consequence of monosaccharides in the chain of **9** established by the ROESY and HMBC spectra were the same as in the glycosides **5** and **6** (Table 9). The comparison of the ^13^C NMR spectra of the compounds **9** and **5** showed the coincidence of the signals corresponding to the monosaccharides from the first to the fourth indicating their identity in these glycosides. The signals of the fifth terminal sugar residue assigned by the ^1^H,^1^H-COSY and 1D TOCSY spectra corresponded to the glucose residue sulfated by C-6 (the signal at δ_C_ 67.9 (C-6 Glc5)). Thus, three sulfate groups were positioned at C-6 of 3-*O*-methylglucose (MeGlc4) and C-6 of two glucose residues (Glc3 and Glc5) in the carbohydrate chain of psolusoside P (**9**). The position of the fourth sulfate group at C-4 Glc5 was established by the comparison of the ^13^C NMR spectra of psolusosides P (**9**) and L (**5**). The signal of C-4 Glc5, deduced by the ^1^H,^1^H-COSY spectrum of **9**, was deshielded to δ_C_ 77.1 due to α-shifting effect of the sulfate group, as compared with the corresponding signal in the ^13^C NMR spectrum of **5** observed at δ_C_ 70.7. Oppositely, the signal of C-5 Glc5 was shielded to δ_C_ 73.7 in the spectrum of **9** due to the β-shifting effect of the sulfate group as compared with the spectrum of **5** (δ_C_ 75.65 (C-5 Glc5)). So, psolusoside P (**9**) is the first case of triterpene glycoside having four sulfate groups, in that two of them were connected to one monosaccharide residue.

The (−)ESI-MS/MS of **9** demonstrated the fragmentation of [M_4Na_ − Na]^−^ ion at *m/z* 1631.4. The peaks of fragment ions were observed at *m/z*: 1265.4 [M_4Na_ − Na − C_6_H_8_O_11_S_2_Na_2_ (Glc(SO_3_Na)_2_)]^−^, 1233.4 [M_4Na_ − Na − C_7_H_12_O_9_SNa (MeGlcSO_3_Na) − NaSO_3_]^−^, 1145.5 [M_4Na_ − Na − C_6_H_9_O_11_S_2_Na_2_ (Glc(SO_3_Na)_2_ − NaSO_4_]^−^, 1089.4 [M_4Na_ − Na − C_7_H_12_O_9_SNa (MeGlcSO_3_Na) − C_6_H_8_O_7_SNa (GlcSO_3_Na)]^−^, 969.4 [M_4Na_ − Na − C_7_H_12_O_9_SNa (MeGlcSO_3_Na) − C_6_H_8_O_7_SNa (GlcSO_3_Na) − NaHSO_4_]^−^, and 943.3 [M_4Na_ − Na − C_7_H_12_O_9_SNa (MeGlcSO_3_Na) − C_6_H_8_O_7_SNa (GlcSO_3_Na) − C_6_H_10_O_4_ (Qui)]^−^ corroborating the structure of carbohydrate chain of psolusoside P (**9**).

All these data indicate that psolusoside P (**9**) is 3*β*-*O*-{6-*O*-sodium-sulfate-3-*O*-methyl-*β*-d-glucopyranosyl-(1→3)-6-*O*-sodium-sulfate-*β*-d-glucopyranosyl-(1→4)-*β*-d-quinovopyranosyl-(1→2)-[4,6-*O*-sodium-disulfate-*β*-d-glucopyranosyl-(1→4)]-*β*-d-xylopyranosyl}-16-ketoholosta-9(11),25-diene.

The molecular formula of psolusoside Q (**10**) was determined to be C_60_H_90_O_40_S_4_Na_4_ from the [M_4Na_ − Na]**^−^** ion peak at *m/z* 1647.3544 (calc. 1647.3590), [M_4Na_ − 2Na]^2**−**^ ion peak at *m/z* 812.1854 (calc. 812.1849), [M_4Na_ − 3Na]^3**−**^ ion peak at *m/z* 533.7944 (calc. 533.7935), and [M_4Na_ − 4Na]^4**−**^ ion peak at *m/z* 394.5989 (calc. 394.5978) in the (−)HR-ESI-MS demonstrating the presence of four sulfate groups. In the ^1^H and ^13^C NMR spectra of the carbohydrate part of psolusoside Q (**10**), five characteristic doublets at δ_H_ 4.61–5.12 (*J* = 6.7–8.4 Hz) and, corresponding to them, signals of anomeric carbons at δ_C_ 101.5–104.8, were indicative of a pentasaccharide chain and *β*-configurations of glycosidic bonds (Table 10). The molecular weights of tetrasulfated psolusosides P (**9**) and Q (**10**) differed by 16 *amu* in HR-ESI-MS that along with the absence of the signals corresponding to the quinovose residue in the NMR spectra of **10** indicated the presence of a glucose residue in the second position of its carbohydrate chain. Actually, the coincidence of the signals of monosaccharide residues from the first to the fourth the ^13^C NMR spectra of psolusosides Q (**10**), N (**7**), and O (**8**) confirmed this supposition. Analysis of the ^1^H,^1^H-COSY, 1D TOCSY, ROESY, and HMBC spectra of psolusoside Q (**10**) showed the same monosaccharide composition and the consequence of monosaccharides in the chain of **10** as in psolusosides N (**7**) and O (**8**) (Table 10). The characteristic signals at δ_C_ 101.5 (C-1 Glc5) and δ_C_ 80.3 (C-2 Glc5) indicated attachment of a sulfate group to C-2 of the fifth residue (Glc5) in the sugar part of **10**. The signal of C-6 Glc5 was assigned by the HSQC spectrum of **10**, demonstrating the correlation of the both doublet at δ_H_ 5.02 (H-6a Glc5) and doublet of doublets at δ_H_ 4.64 (H-6b Glc5) with the corresponding resonance at δ_C_ 67.5 that indicated the presence of an additional sulfate group at C-6 Glc5 in psolusoside Q (**10**). All these data show that psolusoside Q (**10**) has a new carbohydrate chain with four sulfate groups, in that two of them are attached to C-2 and C-6 of the same (Glc5) residue.

The (−)ESI-MS/MS of **10** demonstrated the fragmentation of [M_4Na_ − Na]^−^ ion at *m/z* 1647.4. The peaks of fragment ions were observed at *m/z* 1527.4 [M_4Na_ − Na − NaHSO_4_]^−^, 1281.4 [M_4Na_ − Na − C_6_H_8_O_11_S_2_Na_2_ (Glc(SO_3_Na)_2_)]^−^, 1161.5 [M_4Na_ − Na − C_6_H_8_O_11_S_2_Na_2_ (Glc(SO_3_Na)_2_) − NaHSO_4_]^−^, 1003.4 [M_4Na_ − Na − C_6_H_8_O_11_S_2_Na_2_ (Glc(SO_3_Na)_2_) − C_7_H_11_O_8_SNa (MeGlcSO_3_Na)]^−^, 681.1 [M_4Na_ − Na − C_30_H_43_O_4_ (Agl) − C_6_H_9_O_11_S_2_Na_2_ (Glc(SO_3_Na)_2_) − C_5_H_8_O_4_ (Xyl)]^−^, and 519.0 [M_4Na_ − Na − C_30_H_43_O_4_ (Agl) − C_6_H_9_O_11_S_2_Na_2_ (Glc(SO_3_Na)_2_) − C_5_H_8_O_4_ (Xyl) − C_6_H_10_O_5_ (Glc)]^−^ corroborating the sequence of monosaccharide residues in psolusoside Q (**10**).

All these data indicate that psolusoside Q (**10**) is 3*β*-*O*-{6-*O*-sodium-sulfate-3-*O*-methyl-*β*-d-glucopyranosyl-(1→3)-6-*O*-sodium-sulfate-*β*-d-glucopyranosyl-(1→4)-*β*-d-glucopyranosyl-(1→2)-[2,6-*O*-sodium-disulfate-*β*-d-glucopyranosyl-(1→4)]-*β*-d-xylopyranosyl}-16-ketoholosta-9(11),25-diene.

Thus, highly polar tetrasulfated glycosides are first discovered in sea cucumbers. Although polysulfated polysaccharides are common biopolymers of marine macrophytes and invertebrates, low molecular weight metabolites, containing several sulfate groups are extremely rare. So far, trisulfated natural compounds such as steroid glycosides were found only in sponges [15,16,17] and trisulfated triterpene glycosides, in some representatives of the class Holothuroidea [18,19].

### 2.2. Bioactivity of the Glycosides

The cytotoxic activities of the compounds **1**–**10** as well as known earlier psolusosides G (used as a positive control) and B [12] against mouse erythrocytes (hemolytic activity), the ascite form of mouse Ehrlich ascites carcinoma cells, neuroblastoma Neuro 2A cells, and normal epithelial JB-6 cells are presented in Table 11. The biological effects of the investigated substances were quite different due to the diverse structures of their aglycones and carbohydrate chains. Moreover, hemolytic effects of these compounds were higher than their cytotoxicity against other cells, especially against the Ehrlich ascites carcinoma cells. For instance, psolusoside P (**9**) demonstrated high hemolytic action, but moderate cytotoxicity against Neuro 2A and JB-6 cells and was not active against mouse Ehrlich carcinoma cells (ascite form). The analogic dependency was observed for psolusosides M (**6**) and O (**8**), which were not cytotoxic against all the cell lines except erythrocytes.

Psolusoside L (**5**) was shown to be the most active substance in the series. It has a holostane-type aglycone and pentasaccharide chain with three sulfate groups at C-6 of two glucose and 3-*O*-methylglucose residues. It is very unusual for a glycoside with three sulfate groups to demonstrate high cytotoxic properties, because it is known that sulfate groups attached to the C-6 position of the terminal glucose and 3-*O*-methylglucose residues greatly decrease the activity of pentaosides branched by the second monosaccharide unit (quinovose) sugar chains [3]. Probably, the peculiarities of architecture of a carbohydrate chain of **5** (the branching at C-4 Xyl1) compensate the negative influence of the three sulfate groups.

The activity of psolusoside N (**7**) was slightly lower than that of **5**, due to the presence of a glucose residue as the second unit in the sugar chain instead of the quinovose (in **5**) that is in good accordance with the earlier observations of the glycoside’s SAR [3]. The alteration of the sulfate position attached to the terminal (glucose) residue from C-6 Glc5 to C-2 Glc5 caused the extreme decrease in the activity. This was illustrated by the effects of psolusoside M (**6**) differing from the compound **5** in this character only and demonstrating much lower hemolytic action than **5** and the absence of the activity against other tested cells. The same relationship was observed for psolusosides N (**7**) and O (**8**) differing from each other in the position of the sulfate group in the fifth (Glc5) residue.

The tetrasulfated (at C-6 Glc3, C-6 MeGlc4, C-6 Glc5, and C-4 Glc5) psolusoside P (**9**) demonstrated high hemolytic and moderate cytotoxic action against Neuro-2A and JB-6 cells and was not active against ascites of Ehrlich carcinoma. However, it was much more active than trisulfated psolusoside M (**6**) containing sulfate group at C-2 Glc5. The activity of tetrasulfated psolusoside Q (**10**) was also strongly reduced by the sulfate group attached to C-2 Glc5 as well as by the presence of glucose in the second position of its carbohydrate chain.

Psolusosides B [12], B_1_ (**1**), and B_2_ (**2**) were not active in all the tests due to the presence of non-holostane aglycones in combination with the tetrasaccharide-branched carbohydrate chain sulfated by C-2 of terminal residue (Glc4) attached to C-4 Xyl1. Moreover, psolusosides J (**3**) and K (**4**) with carbohydrate chains with the same architecture and sulfate group at C-2 of the terminal residue (Glc4) were also inactivated despite the presence of holostane aglycones.

## 3. Materials and Methods

### 3.1. General Experimental Procedures

Specific rotation, Perkin-Elmer 343 Polarimeter; NMR, Bruker Avance III 500 (Bruker BioSpin GmbH, Rheinstetten, Germany) (500.13/125.77 MHz) or Avance III 700 Bruker FT-NMR (Bruker BioSpin GmbH, Rheinstetten, Germany) (700.13/176.04 MHz) (^1^H/^13^C) spectrometers were used with tetramethylsilane as the internal standard. ESI MS (positive and negative ion modes), Agilent 6510 Q-TOF apparatus was used with a sample concentration of 0.01 mg/mL. HPLC, Agilent 1100 apparatus with a differential refractometer was used with columns Supelco Ascentis RP-Amide (10 × 250 mm, 5 μm) and Supelco Discovery HS F5-5 (10 × 250 mm, 5 μm).

### 3.2. Animals and Cells

Specimens of the sea cucumber *Psolus fabricii* (family Psolidae; order Dendrochirotida) were collected in the Sea of Okhotsk near Onekotan Island (Kurile Islands). Sampling was performed with a scallop dredge in August–September 1982 at a depth of 100 m during expedition works on fishing seiners “Mekhanik Zhukov” and “Dalarik”. Sea cucumbers were identified by V.S. Levin. Voucher specimens were preserved in the A.V. Zhirmunsky National Scientific Center of Marine Biology, Vladivostok, Russia.

CD-1 mice weighing 18–20 g were purchased from RAMS ‘Stolbovaya’ nursery (Russia) and kept at the animal facility in standard conditions. All experiments were conducted in compliance with all of the rules and international recommendations of the European Convention for the Protection of Vertebrate Animals used for Experimental Studies.

The museum tetraploid strain of murine ascite Ehrlich carcinoma (EAC) cells from the All-Russian Oncology Center (Moscow, Russia) was used. EAC cells were injected into the peritoneal cavity of CD-1 mice. Cells for experimentation were collected 7 days after inoculation. For this purpose, mice were killed by cervical dislocation, and the ascitic fluid containing tumor cells was collected with a syringe. The cells were washed triply by centrifugation at 2000 rpm (450 g) for 10 min in PBS (pH 7.4) followed by resuspension in RPMI-1640 medium containing 8 μg/mL gentamicin (BioloT, Saint Peterburg, Russia). Neuroblastoma Neuro 2A cells were cultured in DMEM medium containing 10% fetal bovine serum (FBS; BioloT, Saint Petersburg, Russia), normal epithelial JB-6 cells were cultured in DMEM medium containing 5% fetal bovine serum (BioloT, Saint Petersburg, Russia), and 1% penicillin/streptomycine (Termo Fisher Scientific (Invitrogen), Waltham, Massachusetts, USA).

### 3.3. Extraction and Isolation

The sea cucumbers (about 800 specimens, average weight of one specimen is about 100 g) were minced and extracted twice with refluxing 60% EtOH. The extract was evaporated to water residuum and lyophilized followed by extraction with CHCl_3_/MeOH (1:1). The obtained extract was evaporated and submitted to the subsequent extraction by EtOAc/H_2_O to remove the lipid fraction. The water layer remaining after this extraction was chromatographed on a Polychrom-1 column (powdered Teflon, Biolar, Olaine, Latvia). The glycosides were eluted with 50% EtOH, evaporated, and subsequently chromatographed on Si gel columns with CHCl_3_/EtOH/H_2_O (100:75:10), (100:100:17), and (100:125:25) as the mobile phase to give subfractions III–VIII containing different groups of glycosides. The continued chromatography on Si gel column of glycosidic sum with CHCl_3_/EtOH/H_2_O (100:125:25) as the mobile phase also gave subfractions IX (602 mg) and X (405 mg). The total weight of all the glycosidic fractions was about 2 g. HPLC of the subfraction VIII on Supelco Ascentis RP-Amide column with CH_3_CN/H_2_O/NH_4_OAc (35/64/1) as the mobile phase gave psolusoside B [6] and other fractions: Ps-B(2) and Ps-B(3). The pure psolusoside B_1_ (**1**) (7.3 mg) was isolated as a result of recromatography of the Ps-B(3) fraction on Discovery HS F5-5 column with MeOH/H_2_O/NH_4_OAc (1 M water solution) (50/49/1) as the mobile phase. Psolusoside B_2_ (**2**) (3.4 mg) was isolated by HPLC of the Ps-B(2) fraction on the same column with MeOH/H_2_O/NH_4_OAc (1 M water solution) (60/38/2) as the mobile phase. The subfraction IX was chromatographed on Supelco Ascentis RP-Amide column with MeOH/H_2_O/NH_4_OAc (1 M water solution) (60/39/1) as the mobile phase to give psolusoside L (**5**) (60 mg) and another subfraction (IXa), that was rechromatographed on Discovery HS F5-5 column with the same solvents in ratio (50/49/1) as the mobile phase to obtain 3.4 mg of psolusoside J (**3**) and 4.8 mg of psolusoside K (**4**). The subfraction X was subjected to HPLC on Supelco Discovery HS F5-5 column with MeOH/H_2_O/NH_4_OAc (1 M water solution) (55/44/1) as mobile phase to give several subsubfractions, rechromatography of which was carried out using different ratios of MeOH/H_2_O/NH_4_OAc (1 M water solution) as mobile phases. The use of the chromatographic system MeOH/H_2_O/NH_4_OAc (60/39/1) resulted in psolusosides M (**6**) (1 mg), N (**7**) (8.8 mg), and P (**9**) (8.5 mg) isolation, the system (52/47/1) gave pure psolusoside O (**8**) (0.6 mg), and the system (50/48.5/1.5) gave 1.4 mg of pure psolusoside Q (**10**).

#### 3.3.1. Psolusoside B_1_ (1)

Colorless powder; [α]_D_^20^ −23 (*c* 0.1, 50% MeOH). NMR: See Table 1 and Appendix A. (−)HR-ESI-MS *m/z*: 1325.4164 (calc. 1325.4185) [M_2Na_ − Na]**^−^**, 651.2157 (calc. 651.2146) [M_2Na_ − 2Na]^2**−**^; (−)ESI-MS/MS *m/z*: 1265.4 [M_2Na_ − Na − CH_3_COOH]^−^, 1145.4 [M_2Na_ − Na − CH_3_COOH − NaHSO_4_]^−^, 1001.4 [M_2Na_ − Na − CH_3_COOH − C_6_H_10_O_8_SNa (GlcSO_3_Na) + H]^−^, 839.3 [M_2Na_ − Na − CH_3_COOH − GlcSO_3_Na − Glc + H]^−^.

#### 3.3.2. Psolusoside B_2_ (2)

Colorless powder; [α]_D_^20^ −18 (*c* 0.1, 50% MeOH). NMR: See Table 2 and Appendix A. (−)HR-ESI-MS *m/z*: 1325.4163 (calc. 1325.4185) [M_Na_ − Na]^−^, 651.2159 (calc. 651.2146) [M_2Na_ − 2Na]^2**−**^; (−)ESI-MS/MS *m/z*: 1265.4 [M_2Na_ − Na − CH_3_COOH]^−^, 1145.4 [M_2Na_ − Na − CH_3_COOH − NaHSO_4_]^−^, 1001.4 [M_2Na_ − Na − CH_3_COOH − C_6_H_10_O_8_SNa (GlcSO_3_Na) + H]^−^, 839.3 [M_2Na_ − Na − CH_3_COOH − GlcSO_3_Na − Glc + H]^−^, 535.1 [M_2Na_ − Na − C_32_H_45_O_6_ (Agl) − C_6_H_10_O_8_SNa (GlcSO_3_Na)]^−^, 403 [M_2Na_ − Na − C_32_H_45_O_6_ (Agl) − C_6_H_10_O_8_SNa (GlcSO_3_Na) − Xyl (C_5_H_8_O_4_)]^−^.

#### 3.3.3. Psolusoside J (3)

Colorless powder; [α]_D_^20^ −17 (*c* 0.1, 50% MeOH). NMR: See Table 3 and Appendix A. (−)HR-ESI-MS *m/z*: 1369.3485 (calc. 1369.3517) [M_3Na_ − Na]**^−^**, 673.1812 (calc. 673.1813) [M_3Na_ − 2Na]^2**−**^, 441.1248 (calc. 441.1244) [M_3Na_ − 3Na]^3**−**^; (−)ESI-MS/MS *m/z*: 1249.4 [M_3Na_ − Na − NaHSO_4_]^−^, 1105.4 [M_3Na_ − Na − C_6_H_9_O_8_SNa (GlcSO_3_Na)]^−^, 1003.4 [M_3Na_ − Na − C_6_H_9_O_8_SNa (GlcSO_3_Na) − NaSO_3_ + H]^−^, 841.4 [M_3Na_ − Na − NaSO_3_ − GlcSO_3_Na − Glc + H]^−^, 403.0 [M_3Na_ − Na − C_30_H_43_O_4_ (Agl) − C_6_H_9_O_11_S_2_Na_2_ (Glc(SO_3_Na)_2_) − Xyl (C_5_H_8_O_4_)]^−^ and 241.0 [M_3Na_ − Na − C_30_H_43_O_4_ (Agl) − C_6_H_9_O_11_S_2_Na_2_ (Glc(SO_3_Na)_2_) − Xyl (C_5_H_8_O_4_) − Glc (C_6_H_10_O_5_)]^−^.

#### 3.3.4. Psolusoside K (4)

Colorless powder; [α]_D_^20^ −16 (*c* 0.1, 50% MeOH). NMR: See Table 4 and Appendix A. (−)HR-ESI-MS *m/z*: 1369.3485 (calc. 1369.3517) [M_3Na_ − Na]**^−^**, 673.1821 (calc. 673.1813) [M_3Na_ − 2Na]^2**−**^, 441.1255 (calc. 441.1244) [M_3Na_ − 3Na]^3**−**^; (−)ESI-MS/MS *m/z*: 1249.4 [M_3Na_ − Na − NaHSO_4_]^−^, 1105.4 [M_3Na_ − Na − C_6_H_9_O_8_SNa (GlcSO_3_Na)]^−^, 1003.4 [M_3Na_ − Na − C_6_H_9_O_8_SNa (GlcSO_3_Na) − NaSO_3_ + H]^−^, 403.0 [M_3Na_ − Na − C_30_H_43_O_4_ (Agl) − C_6_H_9_O_11_S_2_Na_2_ (Glc(SO_3_Na)_2_) − Xyl (C_5_H_8_O_4_)]^−^, 241.0 [M_3Na_ − Na − C_30_H_43_O_4_ (Agl) − C_6_H_9_O_11_S_2_Na_2_ (Glc(SO_3_Na)_2_) − Xyl (C_5_H_8_O_4_) − Glc (C_6_H_10_O_5_)]^−^.

#### 3.3.5. Psolusoside L (5)

Colorless powder; [α]_D_^20^ −35 (*c* 0.1, 50% MeOH). NMR: See Table 5 and Appendix A. (−)HR-ESI-MS *m/z*: 1529.4222 (calc. 1529.4253) [M_3Na_ − Na]**^−^**, 753.2190 (calc. 753.2180) [M_3Na_ − 2Na]^2**−**^, 494.4835 (calc. 494.4823) [M_3Na_ − 3Na]^3**−**^; (−)ESI-MS/MS *m/z*: 1409.5 [M_3Na_ − Na − NaHSO_4_]^−^, 1265.4 [M_3Na_ − Na − C_6_H_9_O_8_SNa (GlcSO_3_Na)]^−^, 1131.5 [M_3Na_ − Na − C_7_H_12_O_9_SNa (MeGlcSO_3_Na) − NaSO_3_]^−^, 665.1 [M_3Na_ − Na − C_30_H_43_O_4_ (Agl) − C_7_H_12_O_9_SNa (MeGlcSO_3_Na) − NaSO_3_]^−^, 519.0 [M_3Na_ − Na − C_30_H_43_O_4_ (Agl) − C_7_H_12_O_9_SNa (MeGlcSO_3_Na) − C_6_H_9_O_7_SNa (GlcSO_3_Na)]^−^.

#### 3.3.6. Psolusoside M (6)

Colorless powder; [α]_D_^20^ −20 (*c* 0.1, 50% MeOH). NMR: See Table 6 and Appendix A. (−)HR-ESI-MS *m/z*: 1529.4273 (calc. 1529.4253) [M_3Na_ − Na]**^−^**, 753.2202 (calc. 753.2180) [M_3Na_ − 2Na]^2**−**^, 494.4844 (calc. 494.4823) [M_3Na_ − 3Na]^3**−**^; (−)ESI-MS/MS *m/z*: 1409.5 [M_3Na_ − Na − NaHSO_4_]^−^, 1265.4 [M_3Na_ − Na − C_6_H_9_O_8_SNa (GlcSO_3_Na)]^−^, 1131.5 [M_3Na_ − Na − C_7_H_12_O_9_SNa (MeGlcSO_3_Na) − NaSO_3_]^−^, 665.1 [M_3Na_ − Na − C_30_H_43_O_4_ (Agl) − C_7_H_12_O_9_SNa (MeGlcSO_3_Na) − NaSO_3_]^−^, 519.0 [M_3Na_ − Na − C_30_H_43_O_4_ (Agl) − C_7_H_12_O_9_SNa (MeGlcSO_3_Na) − C_6_H_9_O_7_SNa (GlcSO_3_Na)]^−^.

#### 3.3.7. Psolusoside N (**7**)

Colorless powder; [α]_D_^20^ −12 (*c* 0.1, 50% MeOH). NMR: See Table 7 and Appendix A. (−)HR-ESI-MS *m/z*: 1545.4171 (calc. 1545.4202) [M_3Na_ − Na]**^−^**, 761.2164 (calc. 761.2155) [M_3Na_ − 2Na]^2**−**^, 499.8151 (calc. 499.8139) [M_3Na_ − 3Na]^3**−**^; (−)ESI-MS/MS *m/z*: 1425.5 [M_3Na_ − Na − NaHSO_4_]^−^, 1281.4 [M_3Na_ − Na − C_6_H_9_O_8_SNa (GlcSO_3_Na)]^−^, 1147.5 [M_3Na_ − Na − C_7_H_12_O_9_SNa (MeGlcSO_3_Na) − NaSO_3_]^−^, 1003.4 [M_3Na_ − Na − C_6_H_9_O_8_SNa (GlcSO_3_Na) − C_7_H_12_O_8_SNa (MeGlcSO_3_Na) + H]^−^, 681.1 [M_3Na_ − Na − C_30_H_43_O_4_ (Agl) − C_7_H_12_O_9_SNa (MeGlcSO_3_Na) − NaSO_3_ + H]^−^, 519.0 [M_3Na_ − Na − C_30_H_43_O_4_ (Agl) − C_7_H_12_O_9_SNa (MeGlcSO_3_Na) − C_6_H_9_O_7_SNa (GlcSO_3_Na)]^−^.

#### 3.3.8. Psolusoside O (**8**)

Colorless powder; [α]_D_^20^ −60 (*c* 0.1, 50% MeOH). NMR: See Table 8 and Appendix A. (−)HR-ESI-MS *m/z*: 1545.4197 (calc. 1545.4202) [M_3Na_ − Na]**^−^**, 761.2171 (calc. 761.2155) [M_3Na_ − 2Na]^2**−**^, 499.8155 (calc. 499.8139) [M_3Na_ − 3Na]^3**−**^; (−)ESI-MS/MS *m/z*: 1425.5 [M_3Na_ − Na − NaHSO_4_]^−^, 1281.4 [M_3Na_ − Na − C_6_H_9_O_8_SNa (GlcSO_3_Na)]^−^, 1161.5 [M_3Na_ − Na − C_6_H_9_O_8_SNa (GlcSO_3_Na) − NaHSO_4_]^−^, 1147.5 [M_3Na_ − Na − C_7_H_12_O_9_SNa (MeGlcSO_3_Na) − NaSO_3_]^−^, 1003.4 [M_3Na_ − Na − C_6_H_9_O_8_SNa (GlcSO_3_Na) − C_7_H_12_O_8_SNa (MeGlcSO_3_Na) + H]^−^, 681.1 [M_3Na_ − Na − C_30_H_43_O_4_ (Agl) − C_7_H_12_O_9_SNa (MeGlcSO_3_Na) − NaSO_3_ + H]^−^, 519.0 [M_3Na_ − Na − C_30_H_43_O_4_ (Agl) − C_7_H_12_O_9_SNa (MeGlcSO_3_Na) − C_6_H_9_O_7_SNa (GlcSO_3_Na)]^−^.

#### 3.3.9. Psolusoside P (**9**)

Colorless powder; [α]_D_^20^ −26 (*c* 0.1, 50% MeOH). NMR: See Table 9 and Appendix A. (−)HR-ESI-MS *m/z*: 1631.3598 (calc. 1631.3641) [M_4Na_ − Na]**^−^**, 804.1879 (calc. 804.1874) [M_4Na_ − 2Na]^2**−**^, 528.4628 (calc. 528.4619) [M_4Na_ − 3Na]^3**−**^, 390.6001 (calc. 390.5991) [M_4Na_ − 4Na]^4**−**^; (−)ESI-MS/MS *m/z*: 1265.4 [M_4Na_ − Na − C_6_H_8_O_11_S_2_Na_2_ (Glc(SO_3_Na)_2_)]^−^, 1233.4 [M_4Na_ − Na − C_7_H_12_O_9_SNa (MeGlcSO_3_Na) − NaSO_3_]^−^, 1145.5 [M_4Na_ − Na − C_6_H_9_O_11_S_2_Na_2_ (Glc(SO_3_Na)_2_ − NaSO_4_]^−^, 1089.4 [M_4Na_ − Na − C_7_H_12_O_9_SNa (MeGlcSO_3_Na) − C_6_H_8_O_7_SNa (GlcSO_3_Na)]^−^, 969.4 [M_4Na_ − Na − C_7_H_12_O_9_SNa (MeGlcSO_3_Na) − C_6_H_8_O_7_SNa (GlcSO_3_Na) − NaHSO_4_]^−^, 943.3 [M_4Na_ − Na − C_7_H_12_O_9_SNa (MeGlcSO_3_Na) − C_6_H_8_O_7_SNa (GlcSO_3_Na) − C_6_H_10_O_4_ (Qui)]^−^.

#### 3.3.10. Psolusoside Q (**10**)

Colorless powder; [α]_D_^20^ −10 (*c* 0.1, 50% MeOH). NMR: See Table 10 and Appendix A. (−)HR-ESI-MS *m/z*: 1647.3544 (calc. 1647.3590) [M_4Na_ − Na]**^−^**, 812.1854 (calc. 812.1849) [M_4Na_ − 2Na]^2**−**^, 533.7944 (calc. 533.7935) [M_4Na_ − 3Na]^3**−**^, 394.5989 (calc. 394.5978) [M_4Na_ − 4Na]^4**−**^; (−)ESI-MS/MS *m/z*: 1527.4 [M_4Na_ − Na − NaHSO_4_]^−^, 1281.4 [M_4Na_ − Na − C_6_H_8_O_11_S_2_Na_2_ (Glc(SO_3_Na)_2_)]^−^, 1161.5 [M_4Na_ − Na − C_6_H_8_O_11_S_2_Na_2_ (Glc(SO_3_Na)_2_) − NaHSO_4_]^−^, 1003.4 [M_4Na_ − Na − C_6_H_8_O_11_S_2_Na_2_ (Glc(SO_3_Na)_2_) − C_7_H_11_O_8_SNa (MeGlcSO_3_Na)]^−^, 681.1 [M_4Na_ − Na − C_30_H_43_O_4_ (Agl) − C_6_H_9_O_11_S_2_Na_2_ (Glc(SO_3_Na)_2_) − C_5_H_8_O_4_ (Xyl)]^−^, 519.0 [M_4Na_ − Na − C_30_H_43_O_4_ (Agl) − C_6_H_9_O_11_S_2_Na_2_ (Glc(SO_3_Na)_2_) − C_5_H_8_O_4_ (Xyl) − C_6_H_10_O_5_ (Glc)]^−^.

### 3.4. Cytotoxic Activity (MTT Assay)

The solutions (20 µL) of tested substances in different concentrations and cell suspension (200 µL) were added in wells of 96-well plates and incubated over night at 37 °C and 5% CO_2_. After incubation the cells were precipitated by centrifugation, 200 µL of medium from each well were collected and 100 µL of pure medium were added. Then 10 µL of MTT (3-(4,5-dimethylthiazol-2-yl)-2,5-diphenyltetrazolium bromide) solution 5 µg/mL (Sigma, St. Louis, MO, USA) were added in each well. The plate was incubated for 4 h, after that 100 µL SDS-HCl were added to each well and the plate was incubated at 37 °C for 4–18 h. Optical density was measured at 570 nm and 630–690 nm. Cytotoxic activity of the substances was calculated as the concentration that caused 50% metabolic cell activity inhibition (IC_50_).

### 3.5. Hemolytic Activity

Blood was taken from CD-1 mice (18–20 g). The mice were anesthetized with diethyl ether, their chests were rapidly opened, and blood was collected in cold (4 °C) 10 mM phosphate-buffered saline, pH 7.4 (PBS) without an anticoagulant. Erythrocytes were washed by centrifugation (2000 rpm) for 5 min, 3 times in PBS using at least 10 vol. of washing solution. Erythrocytes were used at a concentration that provided an optical density of 1.0 at 700 nm for a non-hemolyzed sample. In addition, 20 μL of a water solution of test substance with a fixed concentration was added to a well of a 96-well plate containing 180 μL of the erythrocyte suspension. Erythrocyte suspension was incubated with substances for 24 h at 37 °C. After that, the optical density of the obtained solutions was measured and EC_50_ for hemolytic activity of each compound was calculated. 

## 4. Conclusions

Ten individual compounds **1**–**10**, including di-, tri-, and unprecedented tetrasulfated glycosides were isolated from the sea cucumber *Psolus fabricii*. Psolusosides B_1_ (**1**) and B_2_ (**2**) have disulfated branched by C-4 Xyl1 tetrasaccharide chains identical to that of psolusoside B [12] and non-holostane aglycones with 18(16)-lactone moiety. They differ from other glycosides by their unique structural feature such as 7,8-epoxy-fragment in **2** or by their combination of unusual features such as 7-keto-8,9-ene fragment and 18(16)-lactone in **1**. The compounds **3**–**10** contain common for the sea cucumbers glycosides aglycones, but unique carbohydrate chains. Psolusosides J (**3**) and K (**4**) are characterized by new trisulfated tetrasaccharide-branched chains with the terminal glucose unit sulfated by two positions: by C-2 and C-4 in the compound **3** and by C-2 and C-6 in the compound **4**. Psolusosides L (**5**), M (**6**), and P (**9**) have branched by C-4 Xyl1 pentasaccharide chains with the quinovose as the second sugar unit. These compounds differ from each other by the quantity and positions of sulfate groups in the fifth (Glc5) residue. Psolusoside P (**9**) is tetrasulfated glycoside, containing two sulfate groups at C-4 and C-6 of the same terminal (Glc5) residue. Psolusosides N (**7**), O (**8**), and Q (**10**) have carbohydrate chains with the same architecture, as the glycosides **5**, **6,** and **9** and differ from those by the second monosaccharide residue, which is a glucose instead of a quinovose. Psolusosides N (**7**) and O (**8**) are the structural analogs of psolusosides L (**5**) and M (**6**), correspondingly, having identical sulfate groups positions. Tetrasulfated psolusoside Q (**10**) differs from psolusoside P (**9**) by the positions of sulfation—at C-2 and C-6 of terminal (Glc5) residue. Tetrasulfated glycosides have not ever been found in any natural objects.

The present investigation is conclusive in a series of research concerning the glycosides of the sea cucumber *Psolus fabricii*. Generally, 27 new and 5 known earlier triterpene glycosides have been isolated from this animal. These compounds contain six previously unknown aglycones and 13 novel carbohydrate chains.

The sulfated oligosaccharide moieties predominate in the glycosides of *P. fabricii*. Monosulfated trisaccharide (psolusosides H and H_1_) and linear tetrasaccharide (psolusosides E and F) moieties, disulfated branched tetrasaccharide (psolusosides B, B_1_ (**1**), B_2_ (**2**), and I) or linear tetrasaccharide (psolusosides A and G) carbohydrate chains, trisulfated branched tetrasaccharide (psolusosides J (**3**) and K (**4**)) carbohydrate chains, and finally pentasaccharide trisulfated (psolusosides L (**5**), M (**6**), N (**7**), O (**8**)) and tetrasulfated (psolusosides P (**9**) and Q (**10**)) carbohydrate chains were found in glycosides of *P. fabricii*. The sugar chains also differ from each other by the second monosaccharide unit (quinovose, glucose, or xylose). The most variable structural feature of the carbohydrate chains of the glycosides **1**–**10** is the quantity (one or two) and positions of the sulfate groups in terminal glucose unit, attached to C-4 Xyl1. There are three combinations of such positions of sulfate groups in these residues: C-2 and C-4, C-2 and C-6, or C-4 and C-6. Whereas the single sulfate group bonds only to C-2 or C-6 of terminal glucose unit.

It is interesting to note that diverse groups of psolusosides (A–Q) (a certain group of glycosides consists of substances with the same carbohydrate chain and diverse aglycones) characterized by different structural variability of aglycones. All psolusosides belonging to groups C and D (both containing hexasaccharide non-sulfated sugar chains) have the holostane-type aglycones with 9(11)-double bond, 16-keto-group, and different side chains (5 variants). Psolusosides of the group B contain exclusively non-holostane aglycones with 18(16)-lactone and 7(8)-double bond, completely different from the aglycones of the other groups of psolusosides. These could be explained by their special biological functions in the organism-producer. Four holostane aglycones with 7(8)-, or 9(11)-double bond were found in five glycosides having trisaccharide (psolusosides H and H_1_) or tetrasaccharide-branched carbohydrate chains (psolusosides I, J, K).

Pentaosides (psolusosides L–Q) and tetraosides with linear sugar chains (psolusosides A, E, F, G) contain the same holostane-type aglycone with 9(11)-double bond. It suggests, that linear tetraosides are biosynthetic precursors of pentaosides—psolusosides L (**5**), M (**6**) and P (**9**)—which are biosynthesized via glycosylation and sulfation of psolusosides A, E, and F, correspondingly. Psolusosides N (**7**), O (**8**), and Q (**10**) are formed from psolusoside G through the same processes.

Hence, the biogenetic analysis of the structures of glycosides found in *P. fabricii* showed that carbohydrate chains and aglycones biosynthesis possesses a mosaic (combinatoric) character, which also has some trends.

## Figures and Tables

**Figure 1 marinedrugs-17-00631-f001:**
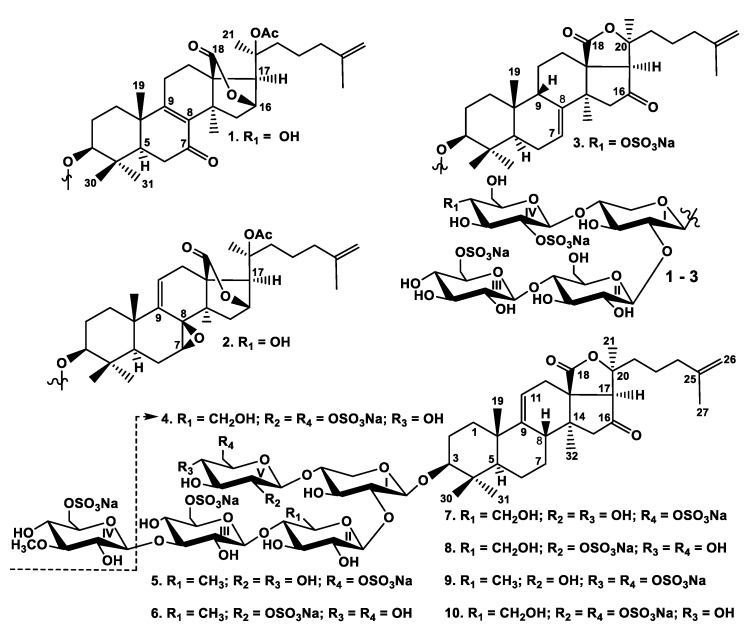
Chemical structures of glycosides isolated from *Psolus fabricii:*
**1**—psolusoside B_1_; **2**—psolusoside B_2_; **3**—psolusoside J; **4**—psolusoside K; **5**—psolusoside L; **6**—psolusoside M; **7**—psolusoside N; **8**—psolusoside O; **9**—psolusoside P; **10**—psolusoside Q.

**Table 1 marinedrugs-17-00631-t001:** ^13^C and ^1^H NMR chemical shifts and HMBC and ROESY correlations of aglycone moiety of psolusoside B_1_ (**1**). *^a^* Recorded at 176.04 MHz in C_5_D_5_N/D_2_O (4/1). *^b^* Recorded at 700.13 MHz in C_5_D_5_N/D_2_O (4/1).

Position	δ_C_ Mult. *^a^*	δ_H_ Mult. (*J* in Hz) *^b^*	HMBC	ROESY
1	34.3 CH_2_	1.71 m		H-11, H-19
		1.24 m		H-3, H-5, H-11
2	26.3 CH_2_	2.00 m		
		1.81 m		H-19, H-30
3	87.7 CH	3.08 dd (4.5; 11.6)	C: 30, C-1 Xyl1	H-5, H-31, H-1 Xyl1
4	39.4 C			
5	50.8 CH	1.54 dd (2.9; 14.5)		H-1, H-3, H-31
6	36.5 CH_2_	2.42 dd (2.6; 15.4)	C: 7, 10	H-31
		2.29 t (15.0)	C: 5, 7	H-19, H-30
7	199.3 C			
8	135.3 C			
9	169.0 C			
10	40.0 C			
11	23.5 CH_2_	2.96 m		H-19
		2.50 dd (9.7; 20.8)		
12	19.1 CH_2_	2.27 m		
		2.14 m		H-32
13	54.9 C			
14	40.1 C			
15	42.5 CH_2_	2.56 d (14.7)	C: 8, 14, 16, 17, 32	
		2.20 d (14.7)	C: 14, 32	H-17, H-32
16	79.9 CH	4.89 brs	C: 13, 14, 18	H-21, H-22, H-23
17	58.8 CH	2.97 s	C: 13, 14, 18, 20, 21, 22	H-15, H-21, H-32
18	179.2 C			
19	18.1 CH_3_	1.14 s	C: 1, 5, 9, 10	H-1, H-2, H-6, H-11, H-30
20	83.8 C			
21	23.3 CH_3_	1.64 s	C: 17, 20, 22	H-16, H-17, H-22
22	37.6 CH_2_	2.23 m		H-17, H-21
		1.83 m		H-16
23	21.1 CH_2_	1.52 m		
		1.45 m		H-17, H-21
24	37.7 CH_2_	1.96 brdd (8.9; 16.2)	C: 22, 23, 25, 26	H-22, H-26, H-27
25	145.4 C			
26	110.7 CH_2_	4.74 brs	C: 24, 27	H-27
27	22.1 CH_3_	1.66 s	C: 24, 25, 26	H-26
30	15.8 CH_3_	0.94 s	C: 3, 4, 5, 31	H-2, H-6, H-19, H-31
31	26.8 CH_3_	1.01 s	C: 3, 4, 5, 30	H-3, H-5, H-6, H-30, H-1 Xyl1
32	27.8 CH_3_	1.41 s	C: 8, 13, 14, 15	H-12, H-15, H-17
OAc	170.9 C			
	21.6 CH_3_	2.09 s	OAc	

**Table 2 marinedrugs-17-00631-t002:** ^13^C and ^1^H NMR chemical shifts and HMBC and ROESY correlations of aglycone moiety of psolusoside B_2_ (**2**). *^a^* Recorded at 176.04 MHz in C_5_D_5_N/D_2_O (4/1). *^b^* Recorded at 700.13 MHz in C_5_D_5_N/D_2_O (4/1).

Position	δ_C_ Mult. *^a^*	δ_H_ Mult. (*J* in Hz) *^b^*	HMBC	ROESY
1	37.1 CH_2_	1.84 m		H-11, H-19
		1.32 m		H-11
2	26.3 CH_2_	1.99 m		
		1.78 m		H-30
3	88.5 CH	3.04 dd (4.3; 11.8)	C: 30, 31, C-1 Xyl1	H-1, H-5, H-31, H-1 Xyl1
4	39.4 C			
5	48.9 CH	1.00 brd (12.1)	C: 4, 6, 10, 19	H-1, H-3, H-31
6	21.0 CH_2_	1.94 m		H-31
		1.73 m	C: 7, 8	H-19, H-30
7	56.2 CH	3.10 d (6.6)	C: 5, 6, 8, 14	H-15, H-32
8	59.6 C			
9	143.2 C			
10	36.7 C			
11	122.7 CH	6.00 brdd (2.6; 5.2)	C: 10, 13	H-1
12	25.2 CH_2_	2.81 dd (5.2; 17.6)	C: 9, 11, 13, 14, 18	H-21
		2.60 brdd (2.3; 17.6)	C: 9, 11, 13, 18	H-21, H-32
13	53.3 C			
14	42.0 C			
15	35.1 CH_2_	1.93 m	C: 14, 16, 17, 32	
		1.50 brdd (2.3; 13.6)	C: 32	H-17, H-32
16	79.3 CH	4.94 brs	C: 13, 14, 18	H-22
17	60.2 CH	3.01 s	C: 13, 14, 18, 20, 21	H-12, H-15, H-21, H-32
18	177.9 C			
19	22.4 CH_3_	1.13 s	C: 1, 5, 9, 10	H-2, H-6, H-30
20	83.9 C			
21	23.4 CH_3_	1.64 s	C: 17, 20, 22	H-12, H-17, H-22
22	37.6 CH_2_	2.24 td (4.6; 13.3)		
		1.85 dd (4.6; 13.7)		H-16
23	21.6 CH_2_	1.52 m	C: 22, 24	
		1.46 m	C: 22, 24	
24	37.7 CH_2_	1.96 m	C: 22, 23, 25	H-22, H-26
25	145.4 C			
26	110.7 CH_2_	4.73 brs	C: 24, 25, 27	H-24, H-27
27	22.1 CH_3_	1.65 s	C: 25, 26	H-26
30	15.8 CH_3_	0.90 s	C: 3, 4, 5, 31	H-2, H-6, H-19
31	27.4 CH_3_	1.09 s	C: 3, 4, 5, 30	H-3, H-5, H-6, H-30, H-1 Xyl1
32	23.9 CH_3_	1.33 s	C: 8, 13, 14, 15	H-7, H-12, H-15, H-17
OAc	170.9 C			
	21.8 CH_3_	2.05 s	OAc	

**Table 3 marinedrugs-17-00631-t003:** ^13^C and ^1^H NMR chemical shifts and HMBC and ROESY correlations of carbohydrate moiety of psolusoside J (**3**). *^a^* Recorded at 176.04 MHz in C_5_D_5_N/D_2_O (4/1). *^b^* Bold = interglycosidic positions. *^c^* Italic = sulphate position. *^d^* Recorded at 700.13 MHz in C_5_D_5_N/D_2_O (4/1). *^e^* Recorded at 500.13 MHz in C_5_D_5_N/D_2_O (4/1). Multiplicity by one-dimensional (1D) TOCSY.

Atom	δ_C_ Mult. ^*a*, *b*, *c*^	δ_H_ Mult. *^d^* (*J* in Hz)	HMBC	ROESY *^e^*
**Xyl1 (1→C-3)**				
1	105.5 CH	4.60 d (7.3)	C: 3	H-3; H-3, 5 Xyl1
2	**81.6 CH**	4.03 t (8.3)	C: 1 Glc2; C: 1, 3 Xyl1	H-1 Glc2
3	75.8 CH	4.24 t (8.8)	C: 2 Xyl1	H-1, 5 Xyl1
4	**79.3 CH**	4.11 m	C: 1 Clc4	H-1 Glc4
5	64.3 CH_2_	4.50 dd (4.9; 11.9)		
		3.81 t (11.2)	C: 1 Xyl1	H-1, 3 Xyl1
**Glc2 (1→2Xyl1)**				
1	104.8 CH	5.12 d (8.1)	C: 2 Xyl1	H-2 Xyl1; H-3, 5 Glc2
2	75.9 CH	3.84 t (8.1)	C: 1, 3 Glc2	
3	76.0 CH	3.98 t (9.5)	C: 2, 4 Glc2	H-1, 5 Glc2
4	**82.8 CH**	3.89 t (9.5)	C: 1 Glc3; C: 5, 6 Glc2	H-1 Glc3
5	76.7 CH	3.71 brd (9.5)		H-1, 3 Glc2
6	62.2 CH_2_	4.31 dd (2.2; 12.0)		
		4.26 dd (7.4; 12.0)		
**Glc3 (1→4Glc2)**				
1	105.3 CH	4.82 d (8.1)	C: 4 Glc2	H-4 Glc2; H-3, 5 Glc3
2	74.9 CH	3.80 t (8.1)	C: 1, 3 Glc3	
3	77.6 CH	4.08 t (9.5)	C: 2, 4 Glc3	
4	71.4 CH	3.92 t (9.5)	C: 3, 6 Glc3	H-6 Glc3
5	76.3 CH	4.03 dd (4,7; 10.1)		H-1, 3 Glc3
6	*68.2* CH_2_	5.01 brd (10.1)		
		4.66 dd (6.1; 10.1)	C: 5 Glc3	
**Glc4 (1→4Xyl1)**				
1	101.7 CH	4.99 d (7.4)	C: 4 Xyl1	H-4 Xyl1; H-3, 5 Glc4
2	*81.2* CH	4.87 t (8.8)	C: 1, 3 Glc4	
3	75.6 CH	4.40 t (8.8)	C: 2, 4 Glc4	H-1, 5 Glc4
4	*77.3* CH	4.90 t (8.8)	C: 3, 5, 6 Glc4	H-6 Glc4
5	76.6 CH	3.84 t (8.8)		H-1, 3 Glc4
6	62.4 CH_2_	4.41 brd (10.2)		
		4.24 dd (5.4; 12.2)		

**Table 4 marinedrugs-17-00631-t004:** ^13^C and ^1^H NMR chemical shifts and HMBC and ROESY correlations of carbohydrate moiety of psolusoside K (**4**). *^a^* Recorded at 176.04 MHz in C_5_D_5_N/D_2_O (4/1). *^b^* Bold = interglycosidic positions. *^c^* Italic = sulphate position. *^d^* Recorded at 700.13 MHz in C_5_D_5_N/D_2_O (4/1). *^e^* Recorded at 500.13 MHz in C_5_D_5_N/D_2_O (4/1). Multiplicity by 1D TOCSY.

Atom	δ_C_ Mult. ^*a*, *b*, *c*^	δ_H_ Mult. *^d^* (*J* in Hz)	HMBC	ROESY *^e^*
**Xyl1 (1→C-3)**				
1	104.7 CH	4.61 d (7.2)	C: 3	H-3; H-3, 5 Xyl1
2	**81.0 CH**	4.00 t (8.7)	C: 1 Glc2; C: 1, 3 Xyl1	H-1 Glc2
3	74.9 CH	4.20 t (8.7)	C: 2, 4 Xyl1	H-1, 5 Xyl1
4	**79.7 CH**	4.01 m	C: 1 Clc4	H-1 Glc4
5	63.6 CH_2_	4.48 dd (5.9; 11.9)	C: 1, 3, 4 Xyl1	
		3.77 t (10.9)		H-1, 3 Xyl1
**Glc2 (1→2Xyl1)**				
1	104.1 CH	5.07 d (7.9)	C: 2 Xyl1	H-2 Xyl1; H-3, 5 Glc2
2	75.2 CH	3.84 t (7.9)	C: 1, 3 Glc2	
3	74.8 CH	4.00 t (8.9)	C: 2, 4 Glc2	H-5 Glc2
4	**82.1 CH**	3.90 t (8.9)	C: 1 Glc3; C: 3, 5, 6 Glc2	H-1 Glc3
5	75.9 CH	3.72 d (9.9)		H-1, 3 Glc2
6	61.2 CH_2_	4.29 dd (4.1; 11.6)		
		4.27 dd (8.4; 11.9)		
**Glc3 (1→4Glc2)**				
1	104.5 CH	4.80 d (8.4)	C: 4 Glc2	H-4 Glc2; H-3, 5 Glc3
2	74.1 CH	3.80 t (8.4)	C: 1, 3 Glc3	
3	76.8 CH	4.08 t (9.2)	C: 2, 4 Glc3	H-1 Glc3
4	70.8 CH	3.88 t (9.2)	C: 3, 5, 6 Glc3	H-6 Glc3
5	75.3 CH	4.03 dd (5.0; 10.1)		H-1 Glc3
6	*67.5* CH_2_	5.00 d (10.9)		
		4.62 dd (7.6; 10.9)	C: 5 Glc3	
**Glc4 (1→4Xyl1)**				
1	101.4 CH	4.90 d (7.6)	C: 4 Xyl1	H-4 Xyl1; H-3, 5 Glc4
2	*80.3* CH	4.72 t (8.4)	C: 1, 3 Glc4	H-4 Glc4
3	76.4 CH	4.26 t (9.2)	C: 2, 4 Glc4	H-1, 5 Glc4
4	70.5 CH	3.92 t (9.2)	C: 3, 5, 6 Glc4	H-2, 6 Glc4
5	75.2 CH	4.04 dd (5.0; 10.9)		H-1, 3 Glc4
6	*67.4* CH_2_	5.00 d (10.1)		
		4.63 dd (6.7; 11.8)	C: 5 Glc4	

**Table 5 marinedrugs-17-00631-t005:** ^13^C and ^1^H NMR chemical shifts and HMBC and ROESY correlations of carbohydrate moiety of psolusoside L (**5**). *^a^* Recorded at 176.04 MHz in C_5_D_5_N/D_2_O (4/1). *^b^* Bold = interglycosidic positions. *^c^* Italic = sulphate position. *^d^* Recorded at 700.13 MHz in C_5_D_5_N/D_2_O (4/1). *^e^* Recorded at 500.13 MHz in C_5_D_5_N/D_2_O (4/1). Multiplicity by 1D TOCSY.

Atom	δ_C_ Mult. ^*a*, *b*, *c*^	δ_H_ Mult. *^d^* (*J* in Hz)	HMBC	ROESY *^e^*
**Xyl1 (1→C-3)**				
1	104.6 CH	4.65 d (7.1)	C: 3; C: 5 Xyl1	H-3; H-5 Xyl1
2	**82.3 CH**	3.89 t (7.9)	C: 1, 3 Xyl1; 1 Qui2	H-1 Qui2; H-4 Xyl1
3	75.0 CH	4.09 t (7.9)	C: 2, 4 Xyl1	H-1, 5 Xyl1
4	**79.2 CH**	4.04 m	C: 3 Xyl1; 1 Glc5	
5	63.5 CH_2_	4.35 dd (5.5; 11.1)	C: 1, 3, 4 Xyl1	
		3.61 dd (9.4; 11.1)	C: 1 Xyl1	H-1 Xyl1
**Qui2 (1→2Xyl1)**				
1	104.7 CH	4.89 d (7.5)	C: 2 Xyl1	H-2 Xyl1; H-5 Qui2
2	75.4 CH	3.84 t (9.0)	C: 1, 3 Qui2	H-4 Qui2
3	74.6 CH	3.91 t (9.0)	C: 2, 4 Qui2	H-1, 5 Qui2
4	**87.1 CH**	3.37 t (8.7)	C: 3, 5 Qui2, 1 Glc3	H-1 Glc3; H-2 Qui2
5	71.4 CH	3.63 dd (6.4; 9.5)		H-1 Qui2
6	17.7 CH_3_	1.59 d (6.4)	C: 4, 5 Qui2	H-4, 5 Qui2
**Glc3 (1→4Qui2)**				
1	104.2 CH	4.73 d (8.1)	C: 4 Qui2	H-4 Qui2; H-3 Glc3
2	73.4 CH	3.84 t (8.1)	C: 1, 3 Glc3	H-4 Glc3
3	**86.5 CH**	4.15 t (8.1)	C: 2, 4 Glc3; 1 MeGlc4	H-1 MeGlc4; H-1 Glc3
4	69.4 CH	3.75 t (9.1)	C: 3, 5, 6 Glc3	H-6 Glc3
5	74.6 CH	4.12 t (9.1)		
6	*67.5* CH_2_	4.98 dd (2.0; 11.0)		
		4.57 dd (7.7; 11.0)	C: 5 Glc3	H-4 Glc3
**MeGlc4 (1→3Glc3)**				
1	104.8 CH	5.16 d (6.9)	C: 3 Glc3	H-3 Glc3; H-3, 5 MeGlc4
2	74.3 CH	3.79 t (8.8)	C: 1, 3 MeGlc4	H-4 MeGlc4
3	86.3 CH	3.64 t (8.8)	C: 2, 4 MeGlc4, OMe	H-1, 5 MeGlc4, OMe
4	69.8 CH	4.01 m	C: 3, 5 MeGlc4	H-2, 6 MeGlc4
5	75.5 CH	4.01 m	C: 4, 6 MeGlc4	H-1, 3 MeGlc4
6	*67.0* CH_2_	4.92 d (10.8)	C: 4, 5 MeGlc4	
		4.75 dd (3.0; 10.8)	C: 5 MeGlc4	
OMe	60.4 CH_3_	3.75 s	C: 3 MeGlc4	
**Glc5 (1→4Xyl1)**				
1	103.4 CH	4.81 d (7.8)	C: 4 Xyl1	H-4 Xyl1; H-3 Glc5
2	73.8 CH	3.81 t (7.8)	C: 1, 3 Glc5	H-4 Glc5
3	76.8 CH	4.10 t (8.8)	C: 2, 4 Glc5	H-1 Glc5
4	70.7 CH	3.92 t (8.8)	C: 3, 5, 6 Glc5	H-2, 6 Glc5
5	75.6 CH	4.06 dd (4.9; 9.8)		H-1 Glc5
6	*67.6* CH_2_	5.02 d (9.8)	C: 4 Glc5	
		4.65 dd (6.9; 11.8)	C: 5 Glc5	H-4 Glc5

**Table 6 marinedrugs-17-00631-t006:** ^13^C and ^1^H NMR chemical shifts and HMBC and ROESY correlations of carbohydrate moiety of psolusoside M (**6**). *^a^* Recorded at 176.04 MHz in C_5_D_5_N/D_2_O (4/1). *^b^* Bold = interglycosidic positions. *^c^* Italic = sulphate position. *^d^* Recorded at 700.13 MHz in C_5_D_5_N/D_2_O (4/1). *^e^* Recorded at 500.13 MHz in C_5_D_5_N/D_2_O (4/1). Multiplicity by 1D TOCSY.

Atom	δ_C_ Mult. ^*a*, *b*, *c*^	δ_H_ Mult. *^d^* (*J* in Hz)	HMBC	ROESY *^e^*
**Xyl1 (1→C-3)**				
1	104.8 CH	4.58 d (7.1)	C: 3	H-3, H-3, 5 Xyl1
2	**82.5 CH**	3.88 t (7.1)	C: 1, 3 Xyl1	H-1 Qui2
3	75.0 CH	4.20 t (8.7)	C: 2, 4 Xyl1	H-1 Xyl1
4	**78.6 CH**	4.13 m		H-1 Glc5
5	63.6 CH_2_	4.48 dd (4.7; 11.9)	C: 3 Xyl1	
		3.77 t (11.9)		H-1 Xyl1
**Qui2 (1→2Xyl1)**				
1	104.5 CH	4.92 d (7.9)	C: 2 Xyl1	H-2 Xyl1; H-5 Qui2
2	75.4 CH	3.85 t (8.7)		H-4 Qui2
3	74.9 CH	3.91 t (8.7)		H-1 Qui2
4	**86.9 CH**	3.39 t (9.6)	C: 1 Glc3; 3, 5 Qui2	H-1 Glc3
5	71.3 CH	3.59 dd (6.0; 9.6)		H-1 Qui2
6	17.7 CH_3_	1.58 d (6.0)		
**Glc3 (1→4Glc2)**				
1	104.1 CH	4.74 d (8.5)	C: 4 Qui2	H-4 Qui2; H-5 Glc3
2	73.4 CH	3.82 t (8.5)		
3	**86.5 CH**	4.13 t (9.3)	C: 4 Glc3; 1 MeGlc4	H-1 MeGlc4; H-1 Glc3
4	69.3 CH	3.75 t (9.3)		
5	74.7 CH	4.11 t (10.1)		H-1 Glc3
6	*67.5* CH_2_	4.99 brd (10.1)		
		4.57 m		
**MeGlc4 (1→3Glc3)**				
1	104.7 CH	5.15 d (7.8)	C: 3 Glc3	H-3 Glc3; H-3, 5 MeGlc4
2	74.3 CH	3.78 t (8.5)	C: 1, 3 MeGlc4	H-4 MeGlc4
3	86.3 CH	3.64 m	C: 4 MeGlc4	H-1, 5 MeGlc4
4	69.8 CH	4.01 m	C: 5 MeGlc4	H-2, 6 MeGlc4
5	75.5 CH	4.01 m		H-1, 3 MeGlc4
6	*67.0* CH_2_	4.92 brd (11.6)		
		4.75 brd (11.6)		H-4 MeGlc4
OMe	60.5 CH_3_	3.76 s	C: 3 MeGlc4	
**Glc5 (1→4Xyl1)**				
1	100.9 CH	4.96 d (7.8)	C: 4 Xyl1	H-4 Xyl1; H-3, 5 Glc5
2	*80.6* CH	4.76 t (7.8)	C: 1 Glc5	H-4 Glc5
3	76.9 CH	4.29 t (8.5)	C: 2, 4 Glc5	H-1, 5 Glc5
4	70.8 CH	3.90 t (8.5)	C: 5 Glc5	
5	77.4 CH	3.87 m		
6	61.8 CH_2_	4.34 brd (10.1)		
		4.01 dd (6.2; 12.4)		

**Table 7 marinedrugs-17-00631-t007:** ^13^C and ^1^H NMR chemical shifts and HMBC and ROESY correlations of carbohydrate moiety of psolusoside N (**7**). *^a^* Recorded at 176.04 MHz in C_5_D_5_N/D_2_O (4/1). *^b^* Bold = interglycosidic positions. *^c^* Italic = sulphate position. *^d^* Recorded at 700.13 MHz in C_5_D_5_N/D_2_O (4/1). *^e^* Recorded at 500.13 MHz in C_5_D_5_N/D_2_O (4/1). Multiplicity by 1D TOCSY.

Atom	δ_C_ Mult.^*a*, *b*, *c*^	δ_H_ Mult.*^d^* (*J* in Hz)	HMBC	ROESY *^e^*
**Xyl1 (1→C-3)**				
1	104.7 CH	4.67 d (6.8)	C: 3	H-3; H-3, 5 Xyl1
2	**81.3 CH**	4.01 t (9.0)	C: 1 Xyl1; 1 Glc2	H-1 Glc2
3	75.0 CH	4.13 t (9.0)	C: 4 Xyl1	H-1, 5 Xyl1
4	**78.8 CH**	4.04 dd (4.5; 9.8)	C: 1 Glc5	H-1 Glc5
5	63.5 CH_2_	4.36 dd (4.5; 10.5)		
		3.63 dd (9.8; 12.0)		H-1 Xyl1
**Glc2 (1→2Xyl1)**				
1	104.3 CH	5.06 d (8.3)	C: 2 Xyl1	H-2 Xyl1; H-3, 5 Glc2
2	75.2 CH	3.87 t (9.0)	C: 1, 3 Glc2	
3	75.2 CH	4.00 t (9.0)	C: 4 Glc2	H-1 Glc2
4	**81.8 CH**	3.95 t (9.0)	C: 3 Glc2, 1 Glc3	H-1 Glc3
5	75.9 CH	3.71 d (9.8)		H-1 Glc2
6	61.0 CH_2_	4.31 dd (3.0; 11.3)		
		4.26 brd (11.3)		
**Glc3 (1→4Glc2)**				
1	103.8 CH	4.84 d (8.3)	C: 4 Glc2	H-4 Glc2; H-3, 5 Glc3
2	73.4 CH	3.83 t (8.3)	C: 1, 3 Glc3	
3	**86.3 CH**	4.10 t (9.0)	C: 4 Glc3; 1 MeGlc4	H-1 MeGlc4; H-1 Glc3
4	69.3 CH	3.75 t (9.0)	C: 5, 6 Glc3	
5	74.8 CH	4.04 dd (6.8; 10.0)		
6	*67.4* CH_2_	4.95 dd (2.3; 10.5)		
		4.57 dd (7.5; 10.5)		
**MeGlc4 (1→3Glc3)**				
1	104.6 CH	5.12 d (8.3)	C: 3 Glc3	H-3 Glc3; H-3, 5 MeGlc4
2	74.3 CH	3.78 t (9.0)	C: 1, 3 MeGlc4	
3	86.3 CH	3.62 t (9.0)	C: 2, 4 MeGlc4, OMe	H-1 MeGlc4, OMe
4	69.7 CH	4.00 t (9.0)	C: 3, 5 MeGlc4	
5	75.5 CH	3.98 m		H-1 MeGlc4
6	*66.9* CH_2_	4.92 dd (2.3; 11.3)		
		4.75 dd (4.5; 11.3)		
OMe	60.4 CH_3_	3.75 s	C: 3 MeGlc4	
**Glc5 (1→4Xyl1)**				
1	103.3 CH	4.81 d (8.3)	C: 4 Xyl1	H-4 Xyl1; H-3 Glc5
2	73.8 CH	3.82 t (9.0)	C: 1, 3 Glc5	
3	76.8 CH	4.10 t (9.0)	C: 2, 4 Glc5	H-1 Glc5
4	70.7 CH	3.94 t (9.0)	C: 3, 6 Glc5	
5	75.6 CH	4.05 dd (3.8; 9.8)		
6	*67.5* CH_2_	5.02 d (9.0)		
		4.67 dd (6.8; 11.3)		

**Table 8 marinedrugs-17-00631-t008:** ^13^C and ^1^H NMR chemical shifts and HMBC and ROESY correlations of carbohydrate moiety of psolusoside O (**8**). *^a^* Recorded at 176.04 MHz in C_5_D_5_N/D_2_O (4/1). *^b^* Bold = interglycosidic positions. *^c^* Italic = sulphate position. *^d^* Recorded at 700.13 MHz in C_5_D_5_N/D_2_O (4/1). *^e^* Recorded at 500.13 MHz in C_5_D_5_N/D_2_O (4/1). Multiplicity by 1D TOCSY.

Atom	δ_C_ Mult. ^*a*, *b*, *c*^	δ_H_ Mult. *^d^* (*J* in Hz)	HMBC	ROESY *^e^*
**Xyl1 (1→C-3)**				
1	104.8 CH	4.60 d (7.1)	C: 3	H-3, H-3, 5 Xyl1
2	**81.1 CH**	4.04 t (8.8)		H-1 Glc2
3	75.1 CH	4.23 t (8.8)	C: 2 Xyl1	
4	**78.7 CH**	4.11 m		H-1 Glc5
5	63.7 CH_2_	4.48 m		
		3.78 t (11.2)		H-1 Xyl1
**Glc2 (1→2Xyl1)**				
1	104.2 CH	5.12 d (8.6)	C: 2 Xyl1	H-2 Xyl1; H-3, 5 Glc2
2	75.2 CH	3.86 t (8.6)	C: 1 Glc2	
3	75.5 CH	3.98 t (8.6)		
4	**81.9 CH**	3.93 t (9.4)		H-1 Glc3
5	75.9 CH	3.70 brd (11.7)		
6	61.2 CH_2_	4.31 brd (11.7)		
		4.27 dd (5.5; 11.7)		
**Glc3 (1→4Glc2)**				
1	103.8 CH	4.86 d (7.8)	C: 4 Glc2	H-4 Glc2; H-3 Glc3
2	73.4 CH	3.81 t (8.6)	C: 1, 3 Glc3	
3	**86.3 CH**	4.08 t (8.6)	C: 2, 4 Glc3; 1 MeGlc4	H-1 MeGlc4; H-1 Glc3
4	69.2 CH	3.77 t (9.4)		
5	74.8 CH	4.04 m		H-1 Glc3
6	*67.3* CH_2_	4.97 brd (9.4)		
		4.58 dd (7.8; 11.7)		
**MeGlc4 (1→3Glc3)**				
1	104.6 CH	5.11 d (7.8)	C: 3 Glc3	H-3 Glc3; H-3, 5 MeGlc4
2	74.3 CH	3.77 t (8.6)	C: 1, 3 MeGlc4	
3	86.4 CH	3.63 t (8.6)	C: 2, 4 MeGlc4, OMe	H-1 MeGlc4
4	69.7 CH	4.02 t (8.6)	C: 5 MeGlc4	
5	75.2 CH	3.99 m		H-1 MeGlc4
6	*66.9* CH_2_	4.92 brd (10.1)		
		4.76 dd (4.7; 12.5)		
OMe	60.4 CH_3_	3.76 s	C: 3 MeGlc4	
**Glc5 (1→4Xyl1)**				
1	101.0 CH	4.94 d (7.0)	C: 4 Xyl1	H-4 Xyl1; H-3, 5 Glc5
2	*80.6* CH	4.76 t (8.6)	C: 1, 3 Glc5	H-4 Glc5
3	76.9 CH	4.30 t (8.6)	C: 2, 4 Glc5	H-1, 5 Glc5
4	70.8 CH	3.91 t (8.6)	C: 5 Glc5	
5	77.4 CH	3.87 m		H-1 Glc5
6	61.8 CH_2_	4.34 dd (2.3; 12.5)		
		4.02 dd (7.0; 12.5)		

**Table 9 marinedrugs-17-00631-t009:** ^13^C and ^1^H NMR chemical shifts and HMBC and ROESY correlations of carbohydrate moiety of psolusoside P (**9**). *^a^* Recorded at 176.04 MHz in C_5_D_5_N/D_2_O (4/1). *^b^* Bold = interglycosidic positions. *^c^* Italic = sulphate position. *^d^* Recorded at 700.13 MHz in C_5_D_5_N/D_2_O (4/1). *^e^* Recorded at 500.13 MHz in C_5_D_5_N/D_2_O (4/1). Multiplicity by 1D TOCSY.

Atom	δ_C_ Mult. ^*a*, *b*, *c*^	δ_H_ Mult. *^d^* (*J* in Hz)	HMBC	ROESY *^e^*
**Xyl1 (1→C-3)**				
1	104.8 CH	4.66 d (7.2)	C: 3	H-3
2	**82.2 CH**	3.87 t (8.8)	C: 1 Xyl1; 1 Qui2	H-1 Qui2
3	75.0 CH	4.08 t (8.8)	C: 2, 4 Xyl1	H-1, 5 Xyl1
4	**79.7 CH**	4.05 m	C: 1 Glc5	H-1 Glc5
5	63.4 CH_2_	4.34 dd (5.6; 11.2)	C: 3 Xyl1	
		3.61 dd (9.6; 12.0)		H-1, 3 Xyl1
**Qui2 (1→2Xyl1)**				
1	104.6 CH	4.87 d (7.8)	C: 2 Xyl1	H-2 Xyl1; H-3, 5 Qui2
2	75.4 CH	3.83 t (7.8)	C: 1, 3 Qui2	H-4 Qui2
3	74.6 CH	3.90 t (8.6)	C: 2, 4 Qui2	H-1, 5 Qui2
4	**87.2 CH**	3.36 t (8.6)	C: 3, 5 Qui2, 1 Glc3	H-1 Glc3, H-2 Qui2
5	71.4 CH	3.62 dd (6.3; 9.4)		H-1 Qui2
6	17.7 CH_3_	1.57 d (5.7)		
**Glc3 (1→4Qui2)**				
1	104.2 CH	4.72 d (8.0)	C: 4 Qui2	H-4 Qui2; H-5 Glc3
2	73.7 CH	3.83 t (8.8)	C: 1, 3 Glc3	H-4 Glc3
3	**86.4 CH**	4.14 t (8.8)	C: 2, 4 Glc3; 1 MeGlc4	H-1 MeGlc4; H-1 Glc3
4	69.4 CH	3.74 t (9.6)	C: 3, 5, 6 Glc3	H-6 Glc3
5	74.5 CH	4.11 t (9.6)		H-1 Glc3
6	*67.5* CH_2_	4.95 dd (2.4; 11.2)		
		4.55 dd (8.0; 11.2)	C: 5 Glc3	H-4 Glc3
**MeGlc4 (1→3Glc3)**				
1	104.7 CH	5.16 d (8.3)	C: 3 Glc3	H-3 Glc3; H-3, 5 MeGlc4
2	74.3 CH	3.78 t (8.3)	C: 1, 3 MeGlc4	H-4 MeGlc4
3	86.5 CH	3.63 t (8.3)	C: 2, 4 MeGlc4, OMe	H-1, 5 MeGlc4, OMe
4	69.8 CH	4.00 m	C: 3, 5 MeGlc4	H-2, 6 MeGlc4
5	75.5 CH	4.01 m	C: 4 MeGlc4	H-1, 3 MeGlc4
6	*67.0* CH_2_	4.92 d (10.6)	C: 4 MeGlc4	
		4.74 dd (4.5; 11.3)		H-4 MeGlc4
OMe	60.5 CH_3_	3.76 s	C: 3 MeGlc4	
**Glc5 (1→4Xyl1)**				
1	103.1 CH	4.79 d (7.7)	C: 4 Xyl1	H-4 Xyl1; H-3, 5 Glc5
2	73.4 CH	3.84 t (7.7)	C: 1, 3 Glc5	H-4 Glc5
3	76.1 CH	4.21 t (8.8)	C: 2, 4 Glc5	H-1 Glc5
4	*77.1* CH	4.68 t (8.8)	C: 3, 5, 6 Glc5	
5	73.7 CH	4.15 dt (9.6; 12.0)		H-1 Glc5
6	*67.9* CH_2_	5.29 dd (2.2; 11.9)		
		4.65 dd (8.8; 11.7)	C: 5 Glc5	

**Table 10 marinedrugs-17-00631-t010:** ^13^C and ^1^H NMR chemical shifts and HMBC and ROESY correlations of carbohydrate moiety of psolusoside Q (**10**). *^a^* Recorded at 176.04 MHz in C_5_D_5_N/D_2_O (4/1). *^b^* Bold = interglycosidic positions. *^c^* Italic = sulphate position. *^d^* Recorded at 700.13 MHz in C_5_D_5_N/D_2_O (4/1). *^e^* Recorded at 500.13 MHz in C_5_D_5_N/D_2_O (4/1). Multiplicity by 1D TOCSY.

Atom	δ_C_ Mult. ^*a*, *b*, *c*^	δ_H_ Mult. *^d^* (*J* in Hz)	HMBC	ROESY *^e^*
**Xyl1 (1→C-3)**				
1	104.8 CH	4.61 d (6.7)	C: 3	H-3, H-3, 5 Xyl1
2	**81.2 CH**	4.01 t (8.9)		H-1 Glc2
3	74.9 CH	4.19 t (8.9)	C: 2 Xyl1	H-1, 5 Xyl1
4	**79.8 CH**	4.02 m	C: 3 Xyl1	H-1 Glc5
5	63.6 CH_2_	4.49 dd (5.2; 11.2)		
		3.77 dd (9.4; 11.2)		H-1 Xyl1
**Glc2 (1→2Xyl1)**				
1	104.2 CH	5.06 d (8.4)	C: 2 Xyl1	H-2 Xyl1; H-3, 5 Glc2
2	75.1 CH	3.84 t (9.2)	C: 3 Glc2	
3	74.6 CH	3.99 t (9.2)	C: 2, 4 Glc2	
4	**81.8 CH**	3.94 t (8.4)	C: 3 Glc2	H-1 Glc3, H-2, 6 Glc2
5	75.9 CH	3.71 d (10.2)		H-1 Glc2
6	60.9 CH_2_	4.29 m		
		4.26 m		
**Glc3 (1→4Glc2)**				
1	103.9 CH	4.84 d (7.4)	C: 4 Glc2	H-4 Glc2; H-3, 5 Glc3
2	73.4 CH	3.82 t (9.2)	C: 1, 3 Glc3	
3	**86.4 CH**	4.10 t (9.2)	C: 2, 4 Glc3; 1 MeGlc4	H-1 MeGlc4; H-1 Glc3
4	69.3 CH	3.76 t (9.2)		
5	74.7 CH	4.04 m		H-1 Glc3
6	*67.4* CH_2_	4.94 d (9.2)		
		4.58 dd (7.4; 12.0)		H-4 Glc3
**MeGlc4 (1→3Glc3)**				
1	104.7 CH	5.12 d (8.3)	C: 3 Glc3	H-3 Glc3; H-3, 5 MeGlc4
2	74.3 CH	3.77 t (8.3)	C: 1, 3 MeGlc4	
3	86.3 CH	3.63 t (8.3)	C: 2, 4 MeGlc4, OMe	H-1, 5 MeGlc4, OMe
4	69.7 CH	4.03 t (9.2)	C: 3, 5, 6 MeGlc4	
5	75.5 CH	3.99 m		H-1, 3 MeGlc4
6	*66.9* CH_2_	4.91 dd (2.1; 11.5)		
		4.77 dd (4.4; 11.1)		
OMe	60.4 CH_3_	3.75 s	C: 3 MeGlc4	
**Glc5 (1→4Xyl1)**				
1	101.5 CH	4.90 d (8.2)	C: 4 Xyl1	H-4 Xyl1; H-3, 5 Glc5
2	*80.3* CH	4.72 t (8.2)	C: 1, 3 Glc5	
3	76.5 CH	4.27 t (8.2)	C: 2, 4 Glc5	H-1, 5 Glc5
4	70.5 CH	3.93 t (9.1)	C: 6 Glc5	
5	75.2 CH	4.05 t (9.1)		H-1 Glc5
6	*67.5* CH_2_	5.02 d (10.4)		
		4.64 dd (6.7; 11.4)		

**Table 11 marinedrugs-17-00631-t011:** The cytotoxic activities of glycosides **1**–**10** and psolusosides B and G (positive control) against mouse erythrocytes, Ehrlich ascites carcinoma cells, mouse neuroblastoma Neuro 2A cells, and normal epithelial JB-6 cells.

Glycoside	Cytotoxicity EC_50_, µM			
	Erythrocytes	Ehrlich Carcinoma	Neuro-2A	JB-6
Psolusoside B	>100.0	>100.0	>100.0	>100.0
Psolusoside B_1_ (**1**)	>100.0	>100.0	>100.0	>100.0
Psolusoside B_2_ (**2**)	>100.0	>100.0	>100.0	>100.0
Psolusoside J (**3**)	>100.0	>100.0	>100.0	>100.0
Psolusoside K (**4**)	>100.0	>100.0	>100.0	>100.0
Psolusoside L (**5**)	2.42	9.73	10.60	7.37
Psolusoside M (**6**)	67.83	>100.0	>100.0	>100.0
Psolusoside N (**7**)	12.37	57.32	13.52	19.94
Psolusoside O (**8**)	34.82	>100.0	>100.0	>100.0
Psolusoside P (**9**)	10.92	>100.0	59.96	56.40
Psolusoside Q (**10**)	>100.0	>100.0	>100.0	>100.0
Psolusoside G	8.86	82.16	35.14	>100.0

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
