# Peer review of "Structures and Bioactivities of Psolusosides B1, B2, J, K, L, M, N, O, P, and Q from the Sea Cucumber Psolus fabricii. The First Finding of Tetrasulfated Marine Low Molecular Weight Metabolites"

_marinedrugs, 2019, doi:10.3390/md17110631_

Round 1

Reviewer 1 Report

The paper is well written, experiments have been correctly planned and methods are sound. The matter of investigation deserves attention for both scientific and technological reasons, and I believe the conclusions are interesting enough to deserve publication.

Since the structural assignments are always quite complex to describe and often do not lead to a fluid reasoning that could be interesting for a broad readership, the paper may result a bit boring in some parts. I admit that this criticism is strictly related to a subjective writing style/preferences and does not alter in any way the value of the work.

A major change I can suggest is related to the need of reporting acquisition parameters for all experiments done. For example, I could not find specific acquisition parameters for most of the NMR spectra acquired. Only in some Figure the acquisition parameters table is reported. I would suggest to complete the paper with this key information.

Paragraph 3.3 reports incomplete information on the quantification of the compounds examined and tested. It is not clear which is the concentration of each compound (mg/g of P. fabricii). This would be particularly relevant, since the biological effect of the investigated substances has been tested.

Author Response

1) The absent and necessary acquisition parameter concerning ROESY was added to the corresponding tables in the manuscript as well as to the corresponding tables in supporting materials. We also have added the information concerning internal standard to the experimental. Note also that all the acquisition parameters are presented in supporting materials in the scans of original spectra.

2) The approximate quantification has been added to the paragraph 3.3. of the experimental.

All the corrections are marked with yellow.

We are very appreciated to the referee for carefull reading of our manuscript and for the useful recommendations.

Reviewer 2 Report

It is described that ten new di-, tri- and tetrasulfated triterpene glycosides, psolusosides B1 (1), B2 (2), J (3), K(4), L (5), M (6), N (7), O (8), P (9) and Q (10), have been isolated from the sea cucumber Psolus fabricii collected in the Sea of Okhotsk near Kurile islands in this paper. Moreover, their structural features were investigated by 2D NMR spectroscopy and HR-ESI mass-spectrometry.

Isolation and to get the structural insights of such natural products are an important research that is directly linked to future drug development. Because, the conformations and assembly structures of organic compounds in solution are important factors that determine the properties of the compound.

In addition, it is known that an amphiphilic compound that combines a hydrophilic part such as a sugar and a fat-soluble part such as a steroid has a very good gellation property. For the reason, the study described in this paper is a very important basic research.

For the reasons described above, I think that this paper should be published in Marine Drugs.

Author Response

Many thanks for your useful recommendations.